# A Stochastic Covariance Shrinkage Approach to Particle Rejuvenation in the Ensemble Transform Particle Filter

Andrey A Popov[1], Amit N Subrahmanya[1], and Adrian Sandu[1]

[1]Computational Science Laboratory, Department of Computer Science, Virginia Tech, 2202 Kraft Drive, Blacksburg, VA, 24060, USA

**Correspondence:** Andrey A Popov (apopov@vt.edu)

**Abstract.** Rejuvenation in particle filters is necessary to prevent the collapse of the weights when the number of particles is insufficient to properly sample the high probability regions of the state space. Rejuvenation is often implemented in a heuristic manner by the addition of random noise that widens the support of the ensemble. This work aims at improving canonical rejuvenation methodology by the introduction of additional prior information obtained from climatological samples; the dynamical particles used for importance sampling are augmented with samples obtained from stochastic covariance shrinkage. A localized variant of the proposed method is developed. Numerical experiments with the Lorenz '63 model show that modified filters significantly improve the analyses for low dynamical ensemble sizes. Furthermore, localization experiments with the Lorenz '96 model show that the proposed methodology is extendable to larger systems.

## 1 Introduction

Ensemble-based data assimilation (Asch et al., 2016; Law et al., 2015; Reich and Cotter, 2015) aims to estimate the state of some dynamical system in a Bayesian framework, and describes the uncertainty through an ensemble of possible states. Describing the distribution of state uncertainty to sufficient accuracy requires very large ensembles, a phenomenon referred to as the curse of dimensionality (Tan et al., 2018; Snyder et al., 2008). Several techniques such as the principle of maximum entropy (Jaynes, 2003) attempt to alleviate this burden by prescribing a distribution constrained by known information. The ensemble Kalman filter (Burgers et al., 1998; Evensen, 1994, 2009), constrains the underlying distributions only by the ensemble mean and covariance; the application of Bayes' rule transforms an assumed prior normal distribution into an assumed posterior normal distribution.

Previous work (Popov et al., 2020) has focused on augmenting the information represented by the ensemble with information derived from covariance shrinkage through a surrogate ensemble in the ensemble transform Kalman filter. In this paper, we extend this idea to the ensemble transport particle filter (ETPF, Reich (2013)). The ETPF transports a given ensemble that represents the posterior distribution using importance sampling (Liu, 2008), to another, equally sampled, ensemble whose moments, in the limit of infinitely many particles, approach the moments of the correct posterior distribution. Like all particle filters, the ETPF is susceptible to weight collapse. Recent attempts to apply particle filters to high dimensional systems (Farchi

and Bocquet, 2018; Van Leeuwen, 2009; Van Leeuwen et al., 2019) have seen some success. However, particle filters are not yet competitive with other state-of-the-art methods such as the ensemble Kalman filter.

This work explores a new approach to particle rejuvenation, which is necessary to prevent weight collapse in particle filters. Rejuvenation in particle filters is a particular type of stochastic regularization (Musso et al., 2001), and is typically implemented in a heuristic manner. Instead of heuristics, our approach makes use of prior information to enrich the ensemble subspace. The new contributions of this work are as follows: (1) we introduce an alternative way of performing particle rejuvenation in the ETPF by incorporating climatological covariance information; (2) we accomplish this by augmenting the dynamical (model) ensemble with synthetic anomalies with optimal scaling, accompanied by a statistically correct estimator; and (3) we show that this rejuvenation method significantly improves the analysis quality for low dynamical ensemble sizes.

This paper is organized as follows. Section 2 reviews the concept of Bayesian inference with the addition of prior information, and its implementation in the context of importance sampling. Section 3 discusses the ensemble transform particle filter and its canonical rejuvenation heuristic. The concept of stochastic covariance shrinkage is proposed in Section 4, and the ETPF is extended to make use of this shrinkage. Numerical experiment results are reported in Section 5. Concluding remarks are drawn in Section 6.

## 2   Optimal coupling with prior information and the ensemble transform particle filter

Bayesian inference (Jaynes, 2003) aims at transforming prior information about the state of a system (represented by the distribution of a random variable $X^{\mathrm{f}}$), additional qualitative and quantitative information ($P$), and information obtained by observing the system ($Y$), into combined posterior information ($X^{\mathrm{a}}$):

$$\pi(X \mid Y, P) = \frac{\pi(Y \mid X, P)\, \pi(X \mid P)}{\pi(Y \mid P)}, \tag{1}$$

where $\pi(X \mid P)$ represents the prior state probability density conditioned by all other relevant information, and $\pi(Y \mid X, P)$ is the observational likelihood conditioned by the forecast $X^{\mathrm{f}}$ and the prior information $P$. Here we consider the finite dimensional case where $X^{\mathrm{f}}, X^{\mathrm{a}} \in \mathbb{R}^n$, $Y \in \mathbb{R}^m$, where the supports of the prior and posterior probability densities are subsets of the respective spaces.

Classical particle filtering (Reich and Cotter, 2015) represents state distributions by collections of particles, i.e., ensembles of samples. Specifically, consider an ensemble of $N^{\mathrm{f}}$ particles $\mathbf{X}^{\mathrm{f}} = [\mathbf{X}^{\mathrm{f}}_1, \ldots \mathbf{X}^{\mathrm{f}}_{N^{\mathrm{f}}}] \in \mathbb{R}^{n \times N^{\mathrm{f}}}$. The prior distribution density is approximated weakly by the corresponding empirical measure,

$$\hat{\pi}(X \mid P) = \sum_{j=1}^{N^{\mathrm{f}}} w^{\mathrm{f}}_j \delta_{X - \mathbf{X}^{\mathrm{f}}_j}, \tag{2}$$

where $w^{\mathrm{f}}_j$ for $1 \le j \le N^{\mathrm{f}}$ are the prior importance weights associated with each particle such that $\sum_i w^{\mathrm{f}}_i = 1$ and $w^{\mathrm{f}}_i > 0$. Similarly, the posterior density is approximated weakly by an empirical measure based on the same sample values (particle

states) but with different posterior importance weights $w_j^{\mathrm{a}}$ for $1 \leq j \leq N^{\mathrm{f}}$:

$$\hat{\pi}(X \mid Y, P) = \sum_{j=1}^{N^{\mathrm{f}}} w_j^{\mathrm{a}} \delta_{X - \mathbf{X}_j^{\mathrm{f}}}, \tag{3}$$

The posterior importance sampling weights are obtained from eq. (1):

$$w_j^{\mathrm{a}} \propto \pi(Y \mid X_j^{\mathrm{f}}, P) \, \pi(X_j^{\mathrm{f}} \mid P) = w_j^{\mathrm{f}} \pi(Y \mid X_j^{\mathrm{f}}, P). \tag{4}$$

The ensemble of weights is denoted by $\boldsymbol{w} = [w_1, \dots w_{N^{\mathrm{f}}}]^T$, and $\boldsymbol{w}^{\mathrm{f}}$ and $\boldsymbol{w}^{\mathrm{a}}$ refer to the forecast and analysis weights respectively. Using (3) and (4) unbiased empirical estimates of the posterior mean and covariance,

$$\bar{\mathbf{x}}^{\mathrm{a}} = \sum_{j=1}^{N^{\mathrm{f}}} w_j^{\mathrm{a}} \mathbf{X}_j^{\mathrm{f}},$$

$$\boldsymbol{\Sigma}_{X^{\mathrm{a}}} = \frac{1}{1 - \boldsymbol{w}^{\mathrm{a},T} \boldsymbol{w}^{\mathrm{a}}} \mathbf{X}^{\mathrm{f}} \left( \mathrm{diag}\left( \boldsymbol{w}^{\mathrm{a}} \right) - \boldsymbol{w}^{\mathrm{a}} \boldsymbol{w}^{\mathrm{a},T} \right) \mathbf{X}^{\mathrm{f},T}, \tag{5}$$

respectively, are obtained by the importance sampling approach (Liu, 2008). The factor in front of the covariance estimate ensures that it is unbiased.

Our goal is to find an analysis ensemble $\mathbf{X}^{\mathrm{a}} \in \mathbb{R}^{n \times N^{\mathrm{a}}}$ of $N^{\mathrm{a}} \leq N^{\mathrm{f}}$ realizations of the random variable $X^{\mathrm{a}}$ that represents the posterior distribution $\pi_{X^{\mathrm{a}}}$ with equal weights. Specifically, the posterior density is approximated weakly by the empirical measure,

$$65 \quad \hat{X}^{\mathrm{a}} \sim \hat{\pi}(X \mid P) = \sum_{j=1}^{N^{\mathrm{a}}} \frac{1}{N^{\mathrm{a}}} \delta_{X - \mathbf{x}_j^{\mathrm{a}}}, \tag{6}$$

where the importance sampling weights are uniform and equal to $1/N^{\mathrm{a}}$ (so as to be equally likely), and $\hat{X}^{\mathrm{a}}$ is the random variable corresponding to this measure which converges weakly in distribution to the exact posterior random variable $X^{\mathrm{a}}$. We impose that the empirical mean (5) is preserved by (6):

$$\bar{\mathbf{x}}^{\mathrm{a}} = \sum_{j=1}^{N^{\mathrm{a}}} \frac{1}{N^{\mathrm{a}}} \mathbf{X}_j^{\mathrm{a}} = \sum_{j=1}^{N^{\mathrm{f}}} w_j^{\mathrm{a}} \mathbf{X}_j^{\mathrm{f}} = \mathbf{X}^{\mathrm{f}} \boldsymbol{w}^{\mathrm{a}}, \tag{7}$$

meaning that the weighted mean of the forecast ensemble is the mean of the analysis ensemble.

The optimal coupling (McCann et al., 1995; Reich and Cotter, 2015) between the prior empirical distribution eq. (2) and the posterior empirical distribution eq. (6), can be defined as an ensemble transformation,

$$\mathbf{X}^{\mathrm{a}} = \mathbf{X}^{\mathrm{f}} \mathbf{T}^*, \tag{8}$$

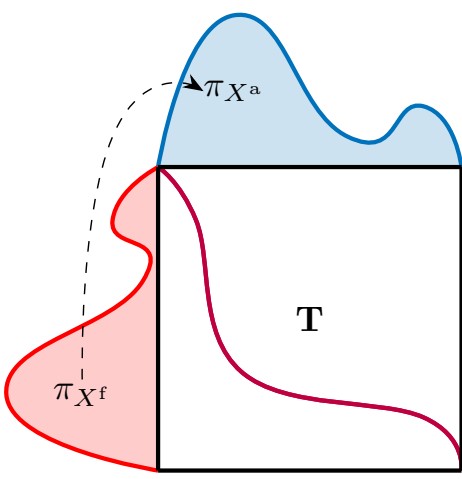

**Figure 1.** A visual representation of the continuous optimal transport procedure. The probability distribution $\pi_{X^{\mathrm{f}}}$ is transported into the distribution $\pi_{X^{\mathrm{a}}}$ through the optimal transport mapping $\mathbf{T}$. The discrete version of the solution is given by (9).

where $\mathbf{T}^* \in \mathbb{R}^{N^{\mathrm{f}} \times N^{\mathrm{a}}}$ is the solution to the optimal transport Monge-Kantorovich problem (Villani, 2003). It is important to see that each element of $\mathbf{T}^*$ is positive. The discrete optimal transportation problem is

$$\mathbf{T}^* = \arg\min_{\mathbf{T}} \sum_{\substack{1 \le j \le N^{\mathrm{f}} \\ 1 \le k \le N^{\mathrm{a}}}} \mathbf{T}_{j,k} \left\| \mathbf{X}_j^{\mathrm{f}} - \mathbf{X}_k^{\mathrm{f}} \right\|_2^2 \tag{9}$$

subject to $\quad \mathbf{T}\mathbf{1}_{N^{\mathrm{a}}} = N^{\mathrm{a}}\boldsymbol{w}^{\mathrm{a}}, \quad \mathbf{T}^T \mathbf{1}_{N^{\mathrm{f}}} = \mathbf{1}_{N^{\mathrm{a}}}, \quad \mathbf{T}_{i,j} \ge 0,$

where the distance measure of squared Euclidean distance is taken for a provably unique solution to the Monge-Kantorovich problem to exist (McCann and Guillen, 2011). The vector of ones of size $q$ is represented by $\mathbf{1}_q$. The problem eq. (9) is a linear programming problem. A visualization of the optimal transport between two continuous distributions is given in Figure 1.

The standard ETPF (Reich, 2013) makes the assumption that the prior and posterior ensemble sizes are the same, $N :=$ $N^{\mathrm{a}} = N^{\mathrm{f}}$ in (9). In the ETPF, most other prior information in $P$ is typically ignored. The discrete optimal transport eq. (8) formulation begets a mapping $\mathbf{X}^{\mathrm{a}} = \Psi_{N^{\mathrm{f}}, N^{\mathrm{a}}}(\mathbf{X}^{\mathrm{f}})$ that, in the limit of ensemble size ($N^{\mathrm{f}} = N^{\mathrm{a}} \to \infty$), converges weakly to a mapping $\Psi$, such that $X^{\mathrm{a}} = \Psi(X^{\mathrm{f}})$ has the exact desired distribution given by eq. (1) (Reich and Cotter, 2015, Theorem 5.19). A second order extension to the ETPF (which we will call 'ETPF2' here) (Acevedo et al., 2017) modifies the optimal transport equation (8) as follows:

$$\mathbf{X}^{\mathrm{a}} = \mathbf{X}^{\mathrm{f}}(\mathbf{T}^* + \mathbf{D}), \tag{10}$$

where the additional term $\mathbf{D}$ is a matrix that ensures that the empirical covariance estimate $\mathbf{\Sigma}_{X^{\mathrm{a}}}$ from (5) is preserved by (6).

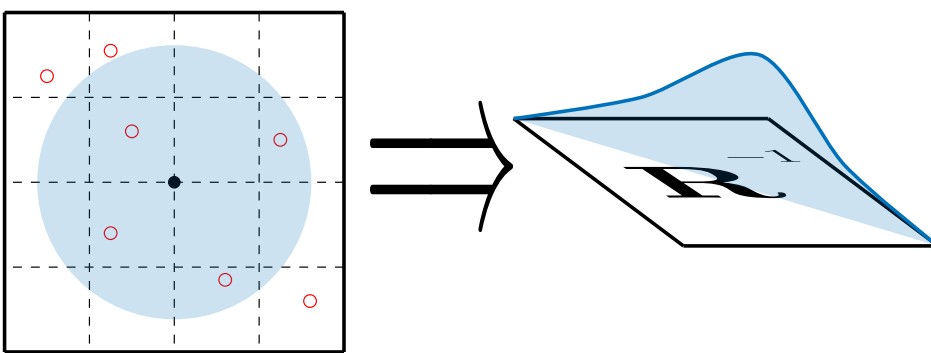

**Figure 2.** A visual representation of $\mathbf{R}$-localization in the ETPF. The left panel represents the localization radius around the $i$th state variable represented by the black dot, with the observations represented by the open red dots. The right panel represents the decorrelation of $\mathbf{R}^{-1}$ along the diagonal.

## 2.1 Localization

In high dimensional geophysical problems, spatial error correlations decrease with increasing spatial distance between states. Due to the undersampled nature of the ensemble, these correlations may not be accurately approximated. Localization allows to strictly enforce the shrinking of correlations between distant states. For localization in the ETPF, we follow the R-localization formulation given in (Reich and Cotter, 2015; Acevedo et al., 2017). Figure 2 provides an illustration of $\mathbf{R}$-localization, with the full procedure described below.

Typically, the observation error distribution is assumed to be unbiased and Gaussian, with the probability density used to compute the weights in (4) for particular realizations of the observation $\mathbf{Y}$, of the state $\mathbf{X}$, and of the prior information $\mathbf{P}$, defined as

$$\pi(\mathbf{Y}|\mathbf{X},\mathbf{P}) = \frac{1}{\sqrt{|2\pi\mathbf{R}|}} e^{-\frac{1}{2}(\mathbf{Y}-\mathcal{H}(\mathbf{X}))^T \mathbf{R}^{-1}(\mathbf{Y}-\mathcal{H}(\mathbf{X}))}, \tag{11}$$

where $\mathcal{H}$ is the observation operator. In this case it can be fully parametrized by the observation error covariance $\mathbf{R} \in \mathbb{R}^{m \times m}$, where $m$ is the number of observations.

We assume that the observations are uncorrelated, making $\mathbf{R} = \mathrm{diag}_j R_{j,j}$ a diagonal matrix. For the $\ell^{th}$ state variable $x^{(\ell)}$, we define the localized observation error covariance matrix $\mathbf{R}_\ell$ via:

$$\mathbf{R}_\ell^{-1} = \mathrm{diag}_{j=1,\ldots,m}\left\{\rho\big(d(\ell,j)/r\big)\right\} \circ \mathbf{R}^{-1}, \tag{12}$$

where $d$ is some distance function defined between the $\ell^{th}$ state space variable and the $j$th observation space variable, $r$ is the localization radius, $\rho$ is a decorrelation function, and $\circ$ stands for the Hadamard element-wise product. In this work we use the Gaspari-Cohn decorrelation function (Gaspari and Cohn, 1999). The localized inverse of the observation error covariance (12) is then used in the generation of the weight ensemble $w_\ell^{\mathrm{a}}$ similar to (4).

A different transform matrix is computed for each state variable. Specifically, consider the ensembles of the $\ell^{th}$ state space variable:

$$\boldsymbol{x}^{\text{f},(\ell)} = \left[x_1^{\text{f},(\ell)}, \quad \ldots, \quad x_{N^{\text{f}}}^{\text{f},(\ell)}\right], \quad \boldsymbol{x}^{\text{a},(\ell)} = \left[x_1^{\text{a},(\ell)}, \quad \ldots, \quad x_{N^{\text{a}}}^{\text{a},(\ell)}\right]. \tag{13}$$

The Monge problem formulation in (9) is replaced by the localized formulation for the $\ell^{th}$ variable,

$$\mathbf{T}_\ell^* = \arg\min_{\mathbf{T}} \sum_{\substack{1 \le j \le N^{\text{f}} \\ 1 \le k \le N^{\text{a}}}} \mathbf{T}_{j,k} \left\|x_j^{\text{f},(\ell)} - x_k^{\text{f},(\ell)}\right\|_2^2$$

$$\text{subject to} \quad \mathbf{T}\mathbf{1}_{N^{\text{a}}} = N^{\text{a}}\boldsymbol{w}_\ell^{\text{a}}, \quad \mathbf{T}^T\mathbf{1}_{N^{\text{f}}} = \mathbf{1}_{N^{\text{a}}}, \quad \mathbf{T}_{i,j} \ge 0, \tag{14}$$

with the analysis ensemble of the $\ell^{th}$ state space variable given by:

$$\boldsymbol{x}^{\text{a},(\ell)} = \boldsymbol{x}^{\text{f},(\ell)}\, \mathbf{T}_\ell^*. \tag{15}$$

## 3 Particle Rejuvenation in the ETPF

Particle and ensemble-based filters often underrepresent uncertainty (Asch et al., 2016) due to the relatively small number of samples when compared to the dimension of the state and data spaces. Over several data assimilation cycles multiple particles start carrying either unimportant or redundant information, which leads to weight collapse or to ensemble degeneracy (Strogatz, 2018). To alleviate these effects, methods such as inflation (Anderson, 2001; Popov and Sandu, 2020), rejuvenation (Reich, 2013), and resampling (Reich and Cotter, 2015; Attia and Sandu, 2015) have been developed.

In order to avoid ensemble collapse, the ETPF employs a particle rejuvenation approach (Acevedo et al., 2017; Reich, 2013; Chustagulprom et al., 2016) that perturbs the analysis ensemble by a random sampling from the ensemble of prior anomalies,

$$\mathbf{X}^{\text{a}} \leftarrow \mathbf{X}^{\text{a}} + \sqrt{\frac{\tau}{N-1}}\mathbf{A}^{\text{f}}\boldsymbol{\eta}\left(\mathbf{I}_N - N^{-1}\mathbf{1}_N\mathbf{1}_N^T\right), \tag{16}$$

where $\boldsymbol{\eta} \sim (\mathcal{N}(0,1))_{N \times N}$ is a matrix of i.i.d. samples from the standard normal distribution of size $N$, the rejuvenation factor $\tau$ (also called the bandwidth parameter) is a hyperparameter that controls the magnitude of the correction, and the ensemble anomalies,

$$\mathbf{A}^{\text{f}} = \mathbf{X}^{\text{f}}\left(\mathbf{I}_N - N^{-1}\mathbf{1}_N\mathbf{1}_N^T\right), \tag{17}$$

are defined as the ensemble of deviations from the sample mean. Of note is the fact that the extra term $\left(\mathbf{I}_N - N^{-1}\mathbf{1}_N\mathbf{1}_N^T\right)$ in (16) ensures that the introduction of the random matrix $\boldsymbol{\eta}$ does not modify the mean of $\mathbf{X}^{\mathrm{a}}$. This is due to the fact that,

$$\left(\mathbf{I}_N - N^{-1}\mathbf{1}_N\mathbf{1}_N^T\right)\mathbf{1}_N = \mathbf{0}_N, \quad \mathbf{1}_N^T\left(\mathbf{I}_N - N^{-1}\mathbf{1}_N\mathbf{1}_N^T\right) = \mathbf{0}_N^T. \tag{18}$$

Note that defining the matrix,

$$\mathbf{B} := \sqrt{\frac{\tau}{N-1}}\left(\mathbf{I}_N - N^{-1}\mathbf{1}_N\mathbf{1}_N^T\right)\boldsymbol{\eta}\left(\mathbf{I}_N - N^{-1}\mathbf{1}_N\mathbf{1}_N^T\right), \tag{19}$$

allows to write the ETPF with standard rejuvenation (16) as follows:

$$\begin{aligned}
\mathbf{X}^{\mathrm{a}} &= \mathbf{X}^{\mathrm{f}}\mathbf{T}^* + \sqrt{\frac{\tau}{N-1}}\mathbf{A}^{\mathrm{f}}\boldsymbol{\eta}\left(\mathbf{I}_N - N^{-1}\mathbf{1}_N\mathbf{1}_N^T\right) \\
&= \mathbf{X}^{\mathrm{f}}\mathbf{T}^* + \sqrt{\frac{\tau}{N-1}}\mathbf{X}^{\mathrm{f}}\left(\mathbf{I}_N - N^{-1}\mathbf{1}_N\mathbf{1}_N^T\right)\boldsymbol{\eta}\left(\mathbf{I}_N - N^{-1}\mathbf{1}_N\mathbf{1}_N^T\right) \\
&= \mathbf{X}^{\mathrm{f}}\left[\mathbf{T}^* + \sqrt{\frac{\tau}{N-1}}\left(\mathbf{I}_N - N^{-1}\mathbf{1}_N\mathbf{1}_N^T\right)\boldsymbol{\eta}\left(\mathbf{I}_N - N^{-1}\mathbf{1}_N\mathbf{1}_N^T\right)\right] \\
&= \mathbf{X}^{\mathrm{f}}\widetilde{\mathbf{T}} \quad \text{with} \quad \widetilde{\mathbf{T}} := \mathbf{T}^* + \mathbf{B}.
\end{aligned} \tag{20}$$

The matrix $\mathbf{B}$ acts as a stochastic perturbation of the optimal transport operator $\mathbf{T}^*$. The choice of $\mathbf{B}$ preserves the linear
constraints of the Monge-Kantorovich problem eq. (9) since $\mathbf{1}_N^T\mathbf{B} = \mathbf{0}_N^T$ and $\mathbf{B}\mathbf{1}_N = \mathbf{0}_N$ due to the property (18). This, of course, immediately calls into question the optimality of the transport for a finite ensemble, as adding this type of $\mathbf{B}$ matrix is perturbing the transport mapping $\widetilde{\mathbf{T}}$ away from the optimum $\mathbf{T}^*$.

## 4  Particle Rejuvenation Through Stochastic Shrinkage

In the context of ensemble methods, covariance shrinkage (Nino-Ruiz and Sandu, 2018, 2015; Ruiz et al., 2014) is used,
similar to other canonical covariance tapering techniques such as inflation (Anderson, 2001; Popov and Sandu, 2020), localization (Anderson, 2012; Hunt et al., 2007; Nino-Ruiz and Sandu, 2017; Nino-Ruiz et al., 2015; Petrie, 2008; Zhang et al., 2010), to enrich the information represented by an undersampled covariance matrix.

From a Bayesian perspective, covariance shrinkage seeks to incorporate additional prior information on error correlations into the analysis, in order to enhance the inference. In many data assimilation models, climatological covariance information
is often available, i.e., it is known prior information. Climatological covariances are typically precomputed or derived from climatological models and are often employed in variational data assimilation (Lorenc et al., 2015).

Following (Popov et al., 2020), we describe the stochastic covariance shrinkage technique. Instead of perturbing the transform matrix as in (20), we instead consider enhancing the dynamic ensemble $\mathbf{X}^{\mathrm{f}} \in \mathbb{R}^{n \times N}$ where $N := N^{\mathrm{f}}$ with an $M$ member synthetic ensemble $\mathcal{X}^{\mathrm{f}} \in \mathbb{R}^{n \times M}$ of samples independent of the dynamical ensemble, where $N^{\mathrm{a}} = N + M$. Each synthetic

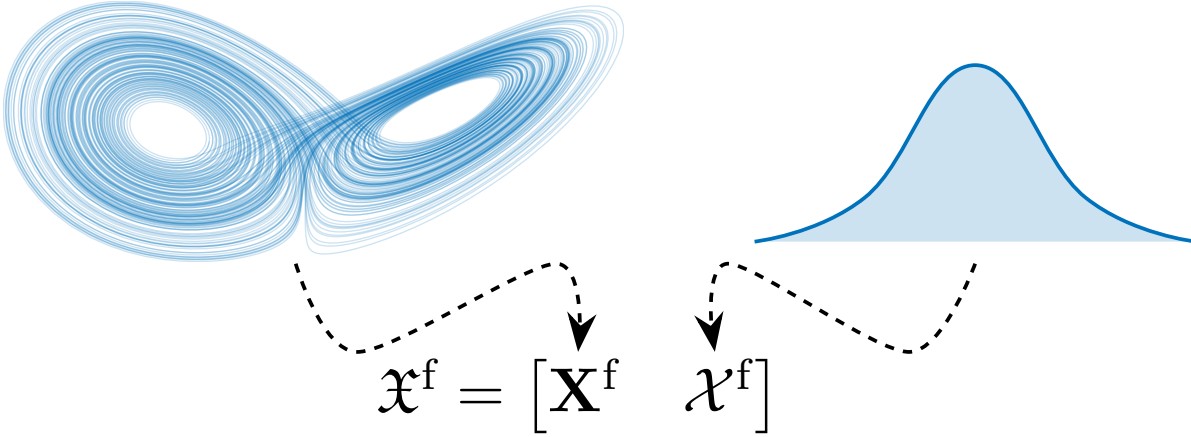

**Figure 3.** A representation of the mixing of the dynamical ensemble $\mathbf{X}^{\mathrm{f}}$ with the synthetic ensemble $\mathcal{X}^{\mathrm{f}}$. The dynamical ensemble comes from the propagation of a dynamical system (here the Lorenz '63 system on the left) while the synthetic ensemble comes from a climatological distribution (here represented by the bell curve on the right).

ensemble member is a biased sample distributed as,

$$\mathfrak{X}^{\mathrm{f}}_{:,i} \sim \pi(X^{\mathrm{f}} \mid P), \tag{21}$$

which is the full distribution of the forecast conditioned by the prior information that we have provided to the algorithm. Note that (21) is not the empirical measure distribution (2), that only has information from the ensemble members, but rather the 'enhanced' distribution that is assumed to contain all the information from the forecast and the prior climatological information

$P$. An illustration of this ensemble combination is shown in Figure 3.

Augmenting the dynamical with the synthetic ensembles leads to the total $N + M$ members ensemble:

$$\mathfrak{X}^{\mathrm{f}} = \begin{bmatrix} \mathbf{X}^{\mathrm{f}} & \mathcal{X}^{\mathrm{f}} \end{bmatrix} \in \mathbb{R}^{n \times (N+M)} \tag{22}$$

with weights $\mathfrak{w}^{\mathrm{f}} = [w^{\mathrm{f}}_1, \dots w^{\mathrm{f}}_{N+M}]$ with respect to $\pi_{P|X^{\mathrm{f}}}$.

Taking $\mathbf{A}^{\mathrm{f}}$ to be the anomalies of the dynamic ensemble (17), and

$$\mathcal{A}^{\mathrm{f}} = \mathcal{X}^{\mathrm{f}} \left( \mathbf{I}_M - M^{-1} \mathbf{1}_M \mathbf{1}_M^T \right), \tag{23}$$

to be the anomalies of the synthetic ensemble, the empirical covariance of the total ensemble can be written as,

$$\boldsymbol{\Sigma}_{\mathfrak{X}^{\mathrm{f}}} = \boldsymbol{\Sigma}_{\mathbf{X}^{\mathrm{f}}} + \boldsymbol{\Sigma}_{\mathcal{X}^{\mathrm{f}}}, \tag{24}$$

where the constituent empirical (unbiased) covariances are defined in terms of the weights

$$\mathbf{\Sigma}_{\mathbf{X}^{\mathrm{f}}} = \sum_{i=1}^{N} w_i^{\mathrm{f}} \frac{N}{N-1} \mathbf{A}_{:,i}^{\mathrm{f}} \mathbf{A}_{:,i}^{\mathrm{f},T}, \quad \mathbf{\Sigma}_{\mathcal{X}^{\mathrm{f}}} = \sum_{i=1}^{M} w_{N+i}^{\mathrm{f}} \frac{M}{M-1} \mathcal{A}_{:,i}^{\mathrm{f}} \mathcal{A}_{:,i}^{\mathrm{f},T}. \tag{25}$$

In the covariance shrinkage approach, to ensure that the sample mean of the augmented ensemble is the same as that of the dynamic ensemble, the synthetic ensemble is constructed with a mean equal to the sample mean of the dynamic ensemble:

$$M^{-1} \mathcal{X}^{\mathrm{f}} \mathbf{1}_M = N^{-1} \mathbf{X}^{\mathrm{f}} \mathbf{1}_N. \tag{26}$$

Thus, constructing the synthetic ensemble only requires sampling the synthetic anomalies. Consider a climatological covariance matrix $\mathcal{P}$. The synthetic anomalies are sampled from some unbiased distribution with covariance $\mu\mathcal{P}$, where $\mu$ is a scaling factor
defined later. In the Gaussian case,

$$\mathcal{A}_{:,i}^{\mathrm{f}} \sim \mathcal{N}(\mathbf{0}_n, \mu\mathcal{P}). \tag{27}$$

An alternate choice of distribution that we explore is the symmetric Laplace distribution (Kozubowski et al., 2013),

$$\mathcal{A}_{:,i}^{\mathrm{f}} \sim \mathcal{L}(\mathbf{0}_n, \mu\mathcal{P}), \tag{28}$$

which is described by the pdf

$$\pi(\mathbf{x}) = \frac{2}{[(2\pi)^n \mu\mathcal{P}]^{\frac{1}{2}}} \left(\frac{\mathbf{x}^T \mathcal{P}^{-1} \mathbf{x}}{2\mu}\right)^{\frac{2-n}{4}} K_{\frac{2-n}{2}} \left(\sqrt{\frac{2\mathbf{x}^T \mathcal{P}^{-1} \mathbf{x}}{\mu}}\right), \tag{29}$$

where $K$ is the modified Bessel function of the second kind (Olver et al., 2010). The choice of Laplace distribution is motivated by robust statistics techniques (Rao et al., 2017). The resulting sampled covariance would therefore be an estimate of the scaled climatological covariance,

$$\mathbf{\Sigma}_{\mathcal{X}^{\mathrm{f}}} \approx \mu\mathcal{P}. \tag{30}$$

**Remark 1.** In order to stay consistent with the mean estimate, the sampled anomalies are replaced with their mean zero counterparts,

$$\mathcal{A}^{\mathrm{f}} \leftarrow \mathcal{A}^{\mathrm{f}} \left(\mathbf{I}_M - M^{-1} \mathbf{1}_M \mathbf{1}_M^T\right). \tag{31}$$

The weights $\mathfrak{w}^{\mathrm{f}}$ are divided into two classes: those that are associated with the dynamic ensemble, and those that are associated with the synthetic ensemble,

$$w_i^{\mathrm{f}} = \begin{cases} 1 - \gamma & 1 \leq i \leq N, \\ \gamma & N+1 \leq i \leq N+M, \end{cases} \tag{32}$$

where the parameter $\gamma$ is known as the covariance shrinkage factor. One choice to calculate $\gamma$ is the Rao-Blackwell Ledoit-Wolf (RBLW) estimator (Chen et al., 2009) (Nino-Ruiz and Sandu, 2017, equation (9)),

$$\gamma_{\mathrm{RBLW}} = \min\left( \left[ \frac{N-2}{N(N+2)} + \frac{(n+1)N-2}{\hat{U}(\mathcal{P}, \boldsymbol{\Sigma}_{\mathbf{X}^{\mathrm{f}}}) N(N+2)(n-1)} \right], 1 \right), \tag{33}$$

where the sphericity factor,

$$\hat{U}(\mathcal{P}, \boldsymbol{\Sigma}_{\mathbf{X}^{\mathrm{f}}}) := \frac{1}{n-1}\left( \frac{n \operatorname{tr}(\mathcal{C}^2)}{\operatorname{tr}^2(\mathcal{C})} - 1 \right), \quad \text{with} \quad \mathcal{C} := \mathcal{P}^{-\frac{1}{2}} \boldsymbol{\Sigma}_{\mathbf{X}^{\mathrm{f}}} \mathcal{P}^{-\frac{1}{2}}, \tag{34}$$

represents the mismatch between the climatological covariance (called the "target" in statistical literature) and sample covariance matrices. Note that if $\boldsymbol{\Sigma}_{\mathbf{X}^{\mathrm{f}}} \to \mathcal{P}$ then $\hat{U}(\mathcal{P}, \boldsymbol{\Sigma}_{\mathbf{X}^{\mathrm{f}}}) \to 0$ and, from (33), $\gamma_{\mathrm{RBLW}} \to 1$. This represents a particular degenerate case whereby the dynamical ensemble is deemed to not be needed, and the climatological distribution is deemed to perfectly represent the forecast.

In this framework (Chen et al., 2009) the scaling parameter for the climatological covariance is defined to be

$$\mu = \frac{\operatorname{tr}(\mathcal{C})}{n}. \tag{35}$$

**Remark 2.** The RBLW estimator (33) makes the assumption that the underlying distribution of the dynamic ensemble is Gaussian. Typically this assumption is violated for dynamical systems of interest.

**Remark 3.** In statistical literature, the target covariance is often taken to be the identity, $\mathcal{P} = \mathbf{I}$, which implies that $\mathcal{C} = \boldsymbol{\Sigma}_{\mathbf{X}^{\mathrm{f}}}$ in (34). The assumption that the target is a climatological covariance is natural generalization in the specific context of sequential data assimilation.

**Remark 4.** The scaling of the target matrix $\mathcal{P}$ is not of any consequence. Let $\widetilde{\mathcal{P}} = \beta \mathcal{P}$ be a scalar scaling of the target matrix, then $\widetilde{\mathcal{C}}_i = \frac{1}{\beta}\mathcal{C}_i$, implying that $\widetilde{\mu}_i = \frac{1}{\beta}\mu_i$, rendering the matrix scaling inconsequential for computing $\mu$. For $\gamma_{\mathrm{RBLW}}$ observe that the trace is linear operation, thus the scaling of $\mathcal{C}$ plays no role in computing $\hat{U}$.

Using the prior weight ensemble determined by (32), the importance sampling weights of the total ensemble $\mathfrak{X}^{\mathrm{f}}$ can be computed using (4), begetting the weight ensemble $\mathfrak{w}^{\mathrm{a}}$. By leveraging this, the resulting analysis ensemble based on prior states and importance sampling weights of $N+M$ states is transported into an equally weighted posterior ensemble of $N$

states through the transformation

$$\mathbf{X}^{\mathrm{a}} = \mathfrak{X}^{\mathrm{f}} \mathbf{T}^*, \tag{36}$$

where the optimal transport matrix $\mathbf{T}^* \in \mathbb{R}^{(N+M)\times N}$ is computed by solving (9).

Recall that in the traditional method of rejuvenation (20), the optimal transport matrix is perturbed randomly into a nearby transport matrix; no new prior information is introduced. We take a fundamentally different approach by incorporating "unseen" prior information derived from a climatological covariance. To this end, before the Monge-Kantorovich problem (9) is solved, we augment of the empirical measure distribution (2) with samples from the climatological distribution, to accommodate the total ensemble (22),

$$\pi(X \mid P) = \sum_{j=1}^{N^{\mathrm{f}}} w_j^{\mathrm{f}} \delta_{X-\mathfrak{X}_j^{\mathrm{f}}}, \tag{37}$$

with $\hat{X}^{\mathrm{f}}$ being the random variable corresponding to this measure, estimating the distribution (21), and the weights (32) coming from the RBLW shrinkage factor (33). In effect we attempt to avoid ensemble collapse by enhancing the empirical measure distribution (37) with new prior information, as opposed to a reweighing of the old prior information. We denote our method as 'FETPF', standing for 'foresight' ETPF, as we believe including prior information in the analysis procedure is a type of foresight. When this procedure is combined with localization as described in section 2.1, we arrive at the Localizaed FETPF, or LFETPF, algorithm.

### 4.1 Convergence of the FETPF

In this section we show that the FETPF reduces to (generalized) variants of the ETPF in two different ways: in the synthetic ensemble limit, and in the synthetic distribution limit.

Assume that the synthetic sample distribution is inexact in the mean and covariance, violating the assumption made in (21). As the dynamical ensemble size $N$ increases, the shrinkage factor $\gamma_{\mathrm{RBLW}}$ in (33) approaches 0. This means that in the limit of an infinite dynamical ensemble, the FETPF reduces to the ETPF.

Assume on the contrary that the synthetic sample distribution is exact, meaning that the climatology produces samples indistinguishable from the forecast, and that the assumption in (21) is fully satisfied. For a finite dynamical ensemble, the shrinkage factor $\gamma_{\mathrm{RBLW}}$ in (33) approaches 1, and the synthetic ensemble is taken as the forecast. This reduces to the ETPF in the case when the dynamical ensemble size is equal to the synthetic ensemble size $N = M$, and should result in an equivalent formulation when $M > N$.

This leaves a gap however, as the shrinkage factor $\gamma_{\mathrm{RBLW}}$ only accounts for the covariance, thus if the synthetic ensemble distribution effectively predicts the covariance, but does not predict the higher order moments well, then the synthetic ensemble will still violate (21) even when it is treated as the forecast. Thus, an ideal shrinkage factor superior to (33) that takes into account more than just the covariance is required, though this is significantly outside the scope of this work.

**Remark 5.** Most ensemble-based methods, including the ETPF, can produce physically unrealistic analysis ensemble members because of the linear nature of the optimal transport matrix. As the ETPF and FETPF perform inference that converges in distribution to the exact analysis distribution, as the dynamical ensemble size $N$ grows, the algorithms produce physically realistic realizations with probability one.

## 4.2 Multiple Climatological Covariance Matrices

It is conceivable that multiple climatological models give rise to multiple climatological covariances, or alternatively multiple candidates for the most 'common' behavior of the model is to be chosen.

Given a collection of target covariances, $\{\mathcal{P}_j\}_{j \in \mathcal{J}}$, we must choose the appropriate covariance from which to sample. We consider the sphericity of the mismatch between the target and forecast covariances eq. (34). Based on the authors' numerical experience, we select the target covariance that corresponds to the highest sphericity of the mismatch:

$$\mathcal{P}^* = \arg\max_{\mathcal{P}_j} \hat{U}(\mathcal{P}_j, \boldsymbol{\Sigma}_{\mathbf{X}^{\mathrm{f}}}), \tag{38}$$

We can justify this choice by realizing that the smaller the sphericity, the closer our samples are to that of canonical rejuvenation techniques. The aim of climatological shrinkage is to introduce unknown information into our inference procedure, thus the target covariance with the highest mismatch introduces the highest amount of outside information.

**Remark 6.** It is also possible to construct 'multi-target' shrinkage estimators (Lancewicki and Aladjem, 2014) that consider all target matrices simultaneously.

## 5 Numerical Experiments

We start with a short introduction to test problems configurations and the numerical experiments setups.

In order to stay in line with other particle rejuvenation techniques, anomaly inflation is used as a heuristic to try and overcome deficiencies in the descriptive power of the synthetic ensemble. Formally,

$$\mathcal{A}^{\mathrm{f}} \leftarrow \alpha \mathcal{A}^{\mathrm{f}}, \tag{39}$$

which is equivalent to assuming an inflated scaling factor $\mu$ in (35). We therefore have two parameters that can be configured in the rejuvenation technique: $M$, the size of our synthetic ensemble, and $\alpha$, the inflation applied to its realizations. It is important that inflation only be applied to the synthetic ensemble, and not the dynamical ensemble, as to not violate the physical constraints of the dynamics.

In our experiments we report the error of the analysis mean with respect to the truth (reference), measured by the spatio-temporal root mean square error (RMSE):

$$\text{RMSE}(\mathbf{x}^t, \bar{\mathbf{x}}^a) = \sqrt{\frac{1}{nT} \sum_{i=1}^{T} \|\mathbf{x}_i^t - \bar{\mathbf{x}}_i^a\|_2^2}, \tag{40}$$

where $T$ stands for the relevant measured timeframe of the experiments.

We now describe the three variable Lorenz '63 model and the 40 variable Lorenz '96 model that are used in the experiments. We use the implementation of both these problems from the ODE test problem suite (Computational Science Laboratory, 2020; Roberts et al., 2019).

## 5.1 Lorenz '63 model

For first set of numerical experiments, we use the Lorenz '63 system (Lorenz, 1963):

$$
\begin{aligned}
x' &= \sigma(y - x), \\
y' &= x(\rho - z) - y, \\
z' &= xy - \beta z,
\end{aligned}
\tag{41}
$$

with chaotic canonical parameter values $\sigma = 10$, $\rho = 28$, and $\beta = 8/3$. We observe the first component, with Gaussian unbiased observation error, with a very large variance of $\mathbf{R} = 8$.

We perform $10,000$ assimilation steps, but discard the first $1,000$ that are used for spinup. The time interval between successive observations is $\Delta t = 0.12$. We perform 20 independent runs and take the mean of the results to obtain an accurate estimate of the expected error. All reported results are for statistically significant differences.

This problem setup is challenging for the ensemble Kalman filter, which does not converge even for larger ensemble sizes. Therefore, this is a relevant test for non-Gaussian algorithms.

## 5.2 Lorenz '63 FETPF analysis results

As discussed previously, the canonical choice for the shrinkage covariance is the identity matrix. It has been the authors' experience that for most dynamical systems this choice is poor. Moreover, the sequential data assimilation problem typically provides ways to calculate climatological approximations to the covariance. We take advantage of such techniques in this paper.

The first type of climatological covariance that we investigate is that of the distribution over the whole manifold of the dynamics. The trace-state normalized,

$$\mathcal{P} \leftarrow \frac{n}{\text{tr}(\mathcal{P})} \mathcal{P} \tag{42}$$

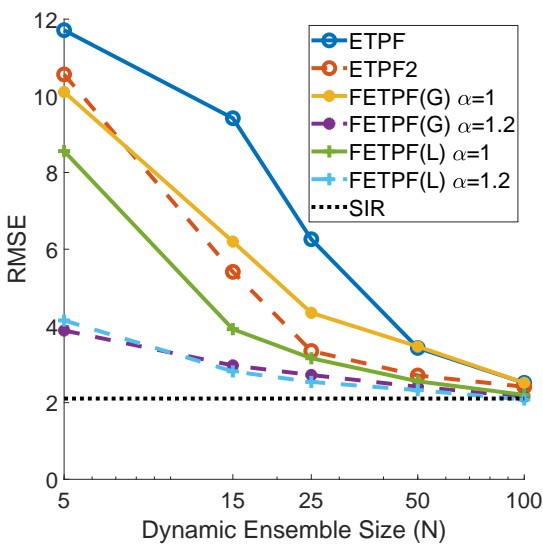

**Figure 4.** For the Lorenz '63 model, analysis RMSE versus dynamic ensemble size ($N$) of the Gaussian (G) and Laplace (L) covariance shrinkage approaches (FETPF) to particle rejuvenation for a synthetic ensemble size of $M = 100$ and for various values of synthetic anomaly inflation, $\alpha$, from (39), with respect to a canonical particle rejuvenation approach for the first order ETPF and the second order ETPF (denoted ETPF2) for an optimal rejuvenation factor $\tau = 0.04$. The target covariance is taken to be (43). The baseline error is denoted by the 'true' SIR filter.

matrix that is obtained by taking the temporal covariance of $50,000$ sample points on the attractor of the canonical Lorenz '63 model is:

$$
\mathcal{P} = \begin{bmatrix} 0.8616 & 0.8618 & -0.0148 \\ 0.8618 & 1.1149 & -0.0035 \\ -0.0148 & -0.0035 & 1.0234 \end{bmatrix}, \tag{43}
$$

with condition number $15.88$.

Our first round of experiments compares the canonical method of rejuvenation in the ETPF and the ETPF2 with a rejuvenation factor of $\tau = 0.04$ in (19) (see Acevedo et al. (2017)) to the stochastic covariance shrinkage technique for both Gaussian and Laplace samples. A dynamic ensemble size $M = 100$, the inflation factors $\alpha \in \{1.0, 1.2\}$, and the target covariance (43) are used. The baseline error is computed using a sequential importance resampling (SIR) filter with an ensemble of $N = 10^5$ particles.

The results for the first round of experiments are shown in Figure 4. From the results, all possible ETPF-based algorithms seem to converge to the SIR baseline for around $N = 100$ particles. The differences between the various algorithms only become apparent at smaller ensemble sizes. As reported in Acevedo et al. (2017), the second-order accurate ETPF2 performs better than the standard ETPF.

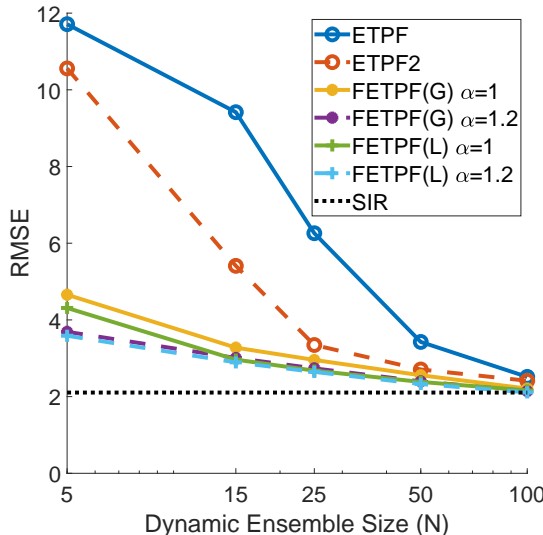

**Figure 5.** For the Lorenz '63 model, analysis RMSE versus dynamic ensemble size ($N$) of the Gaussian (G) and Laplace (L) covariance shrinkage approaches (FETPF) to particle rejuvenation with the multi-target covariances (44) for a synthetic ensemble size of $M = 100$ and for various values of synthetic anomaly inflation ($\alpha$) with respect to a canonical particle rejuvenation approach for the first order ETPF and the second order ETPF (denoted ETPF2) for the rejuvenation factor $\tau = 0.04$. The baseline error is denoted by the 'true' SIR filter.

The FETPF without synthetic inflation performs worse than both the ETPF and the ETPF2 for Gaussian synthetic samples, while it performs better when equipped with Laplacian synthetic samples. When the synthetic samples are inflated with inflation factor $\alpha = 1.2$, the FETPF performs significantly better than all other algorithms.

The second round of experiments uses multiple values of the climatological covariance $\mathcal{P}$. The rest of the setup is identical to the previous experiment. For testing multiple covariances, we run an ETPF with $N = 100$ with $20,000$ evenly spaced state snapshots over a time interval of $2400$ time units and calculate the trace-state normalized forecast covariances. Under a square Frobenius norm distance, we cluster the empirical covariance matrices of the same ensemble at different times using the $k$-means algorithm (Tan et al., 2018) into two clusters. The collection of climatological covariances for the Lorenz '63 thus consists of the centroids of each cluster,

$$\mathcal{P}_1 = \begin{bmatrix} 0.5017 & 0.5524 & -0.4587 \\ 0.5524 & 1.0731 & -0.6723 \\ -0.4587 & -0.6723 & 1.4252 \end{bmatrix}, \quad \mathcal{P}_2 = \begin{bmatrix} 0.5443 & 0.6830 & 0.4330 \\ 0.6830 & 1.2748 & 0.6318 \\ 0.4330 & 0.6318 & 1.1808 \end{bmatrix}, \tag{44}$$

with condition numbers $13.68$ and $16.98$ respectively. As can be seen, the clusters are mainly split by the correlation factors of $z$ with respect to the other variables being positive or negative.

The second round of experiments are reported in Figure 5 using the climatological covariances (44). The analysis of the results is largely similar to the previous experiment, with the only difference being that the FETPF with multiple covariances

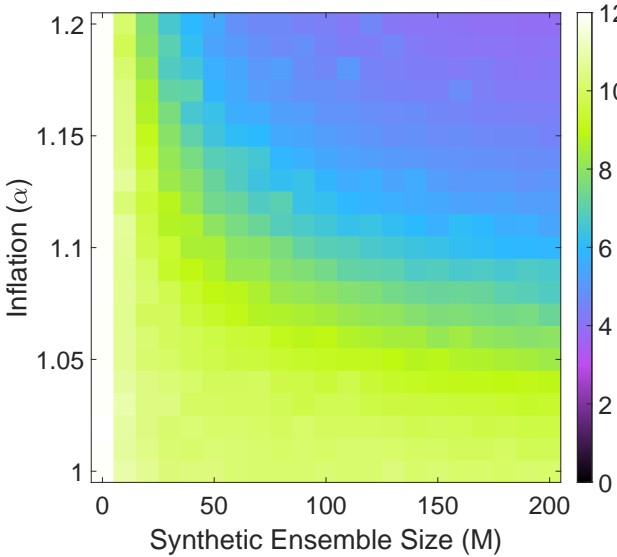

**Figure 6.** For the Lorenz '63 model, analysis RMSE of the covariance shrinkage approach to particle rejuvenation (FETPF) for different values of the synthetic ensemble size $M$ and synthetic inflation factor $\alpha$, and for a dynamical ensemble size of $N = 5$.

does not seem to require synthetic inflation. As the covariance chosen depends on the dynamical ensemble, these results
indicate that a temporally varying climatological distribution might induce an even greater decrease in error.

The results empirically show that supplementing the ensemble with additional synthetic information during assimilation is more effective than randomly perturbing the ensemble post-assimilation, for a small problem. The authors hypothesize that the results point strongly towards the need of intelligently, and adaptively choosing the target covariance matrices, and to the need for better operational calculation of the covariance shrinkage factor $\gamma$.

**5.3   Lorenz '63 parameter search**

Our third round of experiments with the Lorenz '63 system seeks to understand the effect of selecting the two free parameters, i.e., the synthetic ensemble size $M$ and the synthetic ensemble inflation factor $\alpha$. We keep the dynamic ensemble size to a small constant size of $N = 5$, and vary $M$ in the range $[0, 200]$, with $\alpha$ varying in the range $[1, 1.2]$.

Figure 6 shows the spatio-temporal RMSE for various values of $M$ and $\alpha$, with Gaussian synthetic samples using the single
target matrix (43). The results are not surprising. An increase in the synthetic ensemble size $M$ corresponds to a decrease in error. Similarly, an increase in the inflation factor also corresponds to a decrease in error. Furthermore the factors are complementary, meaning that increasing both decreases the error even more significantly.

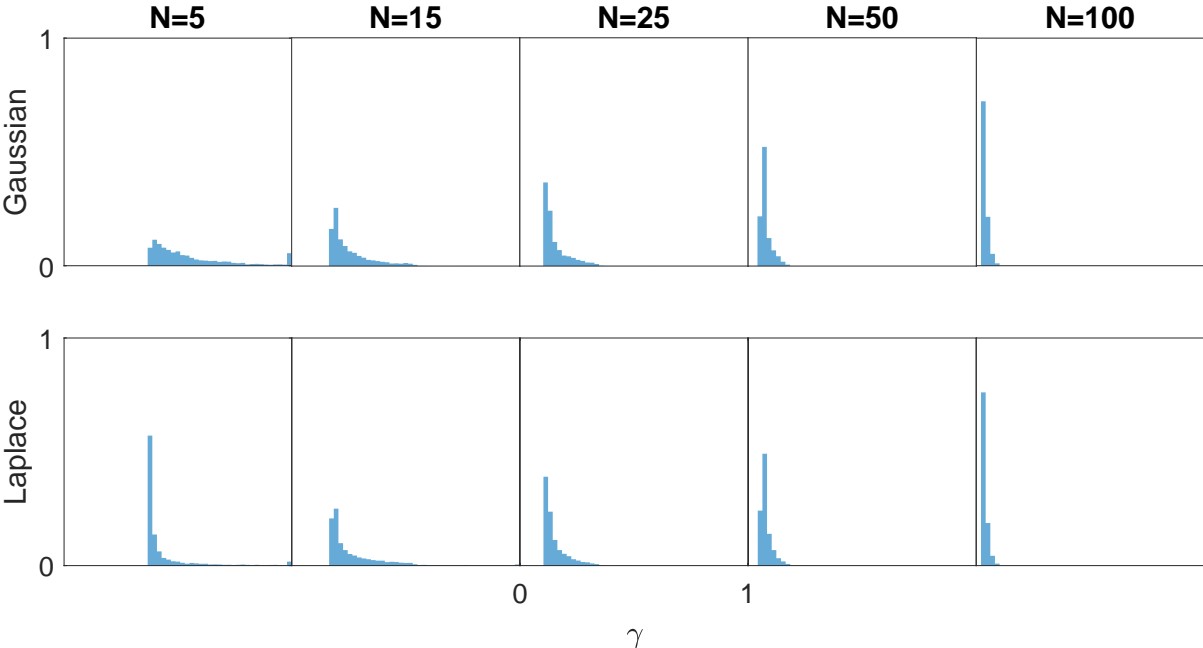

**Figure 7.** The distribution of the $\gamma_{\mathrm{RBLW}}$ parameter for the Lorenz '63 model. Both Gaussian (27) and Laplace (28) assumptions are used for the synthetic ensemble, with no synthetic inflation ($\alpha = 1$).

An interesting effect is that very large synthetic ensemble sizes are required to correspond to a noticeable decrease in error, relative to the dimension of the system. This might pose a challenge when this algorithm is utilized without further corrections such as localization.

### 5.4 Lorenz '63 $\gamma_{\mathrm{RBLW}}$ histograms

Our fourth experiment with the Lorenz '63 equations looks at the distribution of the values of the shrinkage parameter $\gamma_{\mathrm{RBLW}}$ from (33) that is obtained through the assimilation procedure. We test with the Gaussian (27) and Laplace (28) on the climatological distribution with no synthetic inflation (39) ($\alpha = 1$).

Figure 7 shows an approximation to the distributions of $\gamma_{\mathrm{RBLW}}$ for several choices of the dynamical ensemble size $N$ with all other settings kept the same as in the previous experiments. As shown in section 4.1, the shrinkage factor $\gamma_{\mathrm{RBLW}}$ tends towards a distribution that starts resembling a degenerate distribution around zero as $N$ increases. For Gaussian samples, this happens in a smooth fashion, but for Laplace samples something interesting occurs. For $N = 5$, the distribution of $\gamma_{\mathrm{RBLW}}$ is significantly much more skewed towards smaller values, meaning that less confidence is placed in the synthetic ensemble. While the authors do not see a convincing explanation for this behavior, the effect does explain why the Laplace distributed synthetic samples were an improvement over Gaussian samples in 5.2.

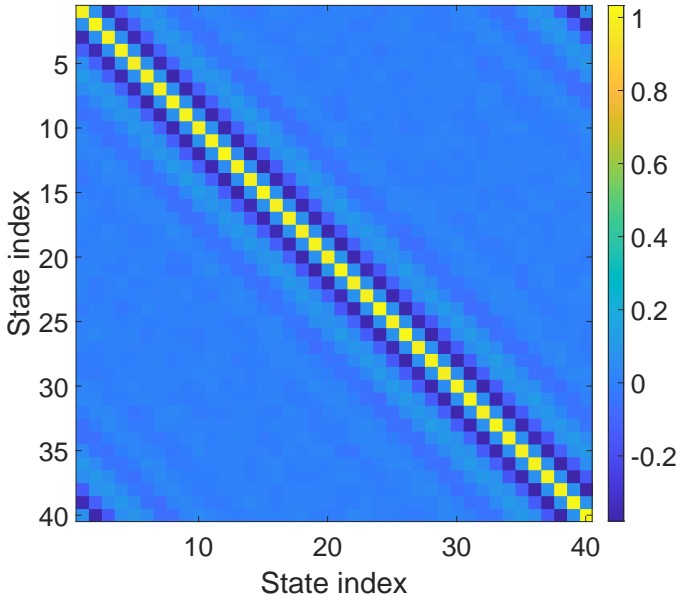

**Figure 8.** Trace-normalized climatological $\mathcal{P}$ covariance matrix used for the Lorenz '96 model.

### 5.5 Lorenz '96 model

For numerical experiments with localization, we use the Lorenz '96 system (Lorenz, 1996; van Kekem, 2018):

$$x_i' = -x_{i-1}\left(x_{i-2} - x_{i+1}\right) - x_i + F, \quad i = 1,\ldots,40, \quad F = 8. \tag{45}$$

with $x_0 = x_{40}$, $x_{-1} = x_{39}$, and $x_{41} = x_1$. The Lorenz '96 system provides a more challenging medium-dimension assimilation scenario. We perform $1,000$ assimilation steps, but discard the first 100 that are used for spinup. The time interval between successive observations is $\Delta t = 0.05$. We perform $4$ independent runs and take the mean of the results to obtain an accurate estimate of the expected error.

     We test with two observation operators. First, we consider a standard linear observation operator

$$\mathcal{H}(\mathbf{x}) = \mathbf{x}, \tag{46}$$

that observes all states with an observation covariance of $\mathbf{R} = \mathbf{I}_{40}$. Second, we use the nonlinear observation operator (Asch et al., 2016),

$$\mathcal{H}(\mathbf{x}) = \frac{\mathbf{x}}{2} \circ \left[1 + \left(\frac{|\mathbf{x}|}{10}\right)^{\circ(\omega-1)}\right], \tag{47}$$

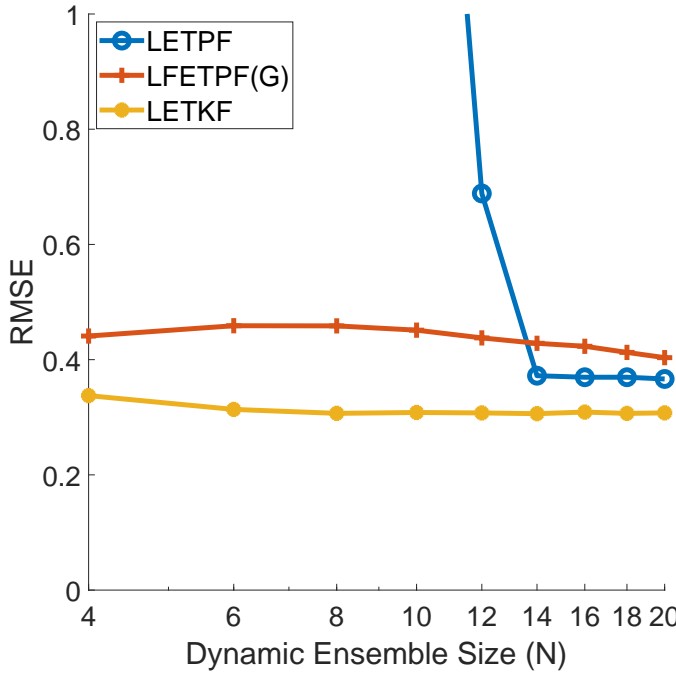

**Figure 9.** For the Lorenz '96 model with the linear observation operator (46): analysis RMSE versus dynamic ensemble size ($N$) of the LETPF with rejuvenation factor $\tau = 0.2$ versus the Gaussian (G) localized covariance shrinkage approach (LFETPF) to particle rejuvenation for a synthetic ensemble size of $M = 100$ and for synthetic anomaly inflation, $\alpha = 1.05$, versus the LETKF (Hunt et al., 2007) method with inflation $\alpha = 1.05$.

where ○ stands for element-wise operations (multiplication and exponentiation, and $|\cdot|$ stands for element-wise absolute value).

that observe all states through a non-linear fashion with the observation covariance $\mathbf{R} = \mathbf{I}_{40}$. We set the control parameter to $\omega = 5$ for a moderately non-linear system.

The matrix $\mathcal{P}$ is computed in a similar way as (43) for the Lorenz '63 Model, and is shown in fig. 8.

For localization, we take the Gaspari-Cohn Gaspari and Cohn (1999) decorrelation function. As observed in Farchi and Bocquet (2018), a relatively small localization radius is needed for the LETPF to converge. We take a radius of $r = 1$ variable,

with the internal parameter of $\theta = \frac{3\sqrt{2\pi}}{7 - 4\log(2)} \approx 1.77884$ to more closely match the standard Gaussian localization function (see Petrie (2008) for more details on the internal parameter). This effectively means that three variables on either size are taken into consideration for every state variable.

### 5.6   Lorenz '96 localization results

For the Lorenz '96 experiments we aim to compare LFETPF to the LETPF, and to the Localized ensemble transform Kalman

filter (LETKF). We set the synthetic ensemble size to a constant $M = 100$ and the synthetic ensemble inflation factor to $\alpha = 1.05$. The inflation factor $\alpha = 1.05$ is also used in the LETKF. We vary the dynamical ensemble size in the range $N \in [2, 20]$,

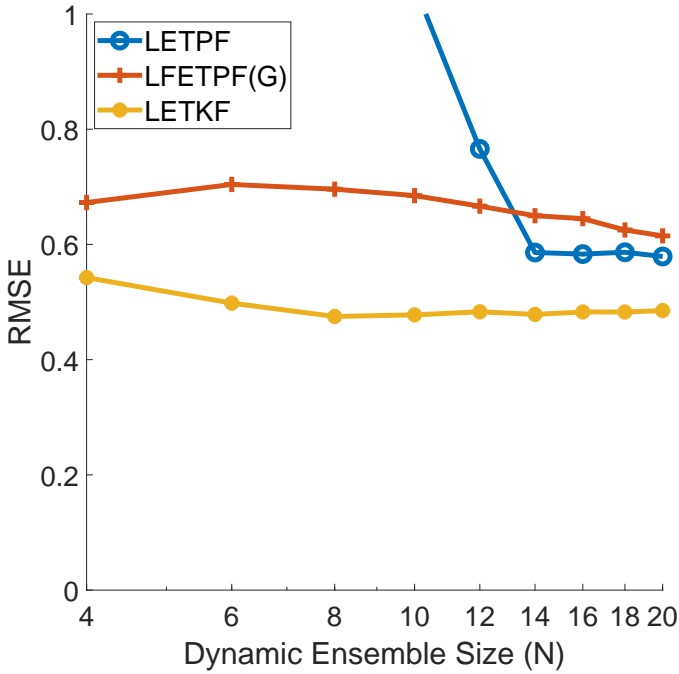

**Figure 10.** For the Lorenz '96 model with the nonlinear observation operator (47): analysis RMSE versus dynamic ensemble size ($N$) of the LETPF with rejuvenation factor $\tau = 0.2$ versus the Gaussian (G) localized covariance shrinkage approach (LFETPF) to particle rejuvenation for a synthetic ensemble size of $M = 100$ and for synthetic anomaly inflation, $\alpha = 1.05$, versus the LETKF (Hunt et al., 2007) method with inflation $\alpha = 1.05$.

and plot the spatio-temporal analysis RMSE over the time interval after spinup. For the LETPF, a constant high rejuvenation value of $\tau = 0.2$ is utilized in order to ensure convergence.

For the linear observation operator (46), the results of this experiment can be seen in Figure 9. The proposed LFETPF did converge, for a dynamical ensemble size of as little as $N = 4$ as compared to the LETPF which required a minimum ensemble size of $N = 14$, in line with the number of positive Lyapunov modes of the system. The proposed LFETPF performed slightly worse than the LETPF, which we suspect would be rectified by a larger choice of $M$. Both the particle filters performed worse than the state-of-the-art LETKF.

For the non-linear observation operator (47) the results are plotted in Figure 10. Similar to the linear observation operator, both the LETKF and LFETPF require very few ensemble members to converge, with the LETPF again requiring around $N = 14$ ensemble members. The LETKF still beats both particle filters even in this non-Gaussian setting. These results show that the R-localization approach used for the ETPF is likely breaking many of the nice non-Gaussian features of the algorithm, as the algorithm should theoretical perform significantly better than a Kalman filtering based approach in this setting.

## 6  Conclusions

This paper introduces a stochastic covariance shrinkage-based particle rejuvenation technique for the ensemble transport particle filter. Instead of incorporating synthetic noise as an attempt to regularize the distribution of the ensemble, we attempt to incorporate an ensemble derived from some known prior information. This is done through the use of synthetic anomalies. These synthetic anomalies are sampled from from any chosen distribution family, such that they are consistent with the climatological covariance information. We provide a philosophical justification for why we believe our approach is more in line with the assumptions underlying Bayesian inference.

Numerical experiments with the simple three variable Lorenz system show that the use of climatological prior information to perform rejuvenation leads to reduced analyses errors than the typical rejuvenation approach. Additionally, the FETPF methodology seems to be much more stable for smaller dynamical ensemble sizes than the original rejuvenation approach. This leads us to believe that the stochastic shrinkage approach augments the original ensemble in a meaningful way.

Numerical experiments with localization techniques show that the the LFETPF is comparable in performance to the LETPF, however a large synthetic ensemble size is likely needed. Future work combining the LFETPF and the LFETKF Popov et al. (2020) in a hybrid filtering approach Acevedo et al. (2017) might provide for a happy medium between the LFETPF and the LFETKF.

One limitation of this work is the focus on synthetic Gaussian samples. Methods such as generative adversarial networks (Aggarwal et al., 2018) (known as GANs) could be utilized to give more complex synthetic samples that mimic "true" samples in meaningful ways. Another related limitation is our focus solely on the RBLW (Chen et al., 2009) shrinkage factor. While it has been the authors' experience that other factors do not perform as well in this setting, further research into building factors specifically tailored for ensemble-based data assimilation methods is warranted.

By far the largest limitation of this work is common to research on particle filters: the large dimensional setting. While localization methods have begun the foray of particle filters into the medium dimensional setting, alternatives to R-localization in the ETPF, coupled with stochastic shrinkage may provide an avenue to a higher-dimensional setting.

It is the authors' belief that future research into all the limitations that have been identified might significantly improve the performance of the FETPF, and create a method that can be applied to operational problems.

### Acknowledgements

This work was supported by DOE through award ASCR DE-SC0021313, by NSF through award CDS&E–MSS 1953113, and by the Computational Science Laboratory at Virginia Tech.

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
