# Peer review of "A Stochastic Covariance Shrinkage Approach to Particle Rejuvenation in the Ensemble Transform Particle Filter"

_Nonlinear Processes in Geophysics, 2021_

## Referee Comment (RC1)

**A stochastic covariance shrinkage approach to particle rejuvenation in the ensemble transform particle filter – Review report**

13th October 2021

In this article, the authors derive a new particle filter algorithm. The algorithm starts by adding some additional members to the ensemble. These additional members are drawn from a normal distribution with a static covariance matrix. The algorithm then uses the ensemble transform algorithm of Reich (2013) to construct the analysis ensemble. The whole idea of this method is to replace the post-analysis regularisation process (called particle rejuvenation in this article) which is usually necessary with particle filtering. The algorithm is finally illustrated using three test series of twin simulations with the 3-variable Lorenz system.

Overall the paper is well written and easy to follow. The presentation of the method is appropriate and understandable. However, I have the impression that several aspects could be improved and that a key methodological aspect is avoided. The presentation of the experiments is correct as well, but a few experiments with a 3-variable system is not enough to make a convincing illustration. In addition, the experimental results are barely discussed and the conclusion is much too short.

**1 General comments**

**1.1 Notation**

Throughout the manuscript, the notation is inconsistent. For example, in Eq. (2) the argument if $X|P$ while in Eq. (3) it is $x|y, p$. Using a consistent notation would really make the manuscript clearer and hence help the reader. Furthermore, I strongly recommend to follow the usual conventions of the data assimilation community (which, if I am not mistaken, also coincide with the journal conventions):

- bold face uppercase for matrices (ex $\mathbf{M}$);

- bold face lowercase italic for vectors (ex $\boldsymbol{v}$);

- lowercase italic or greek letters for scalar quantities (ex $n$ or $\alpha$);

- uppercase italic for sizes (ex $N$).

**1.2 Regularisation or particle rejuvenation**

The entire method derived by the authors is designed as a sort of extension of the ETPF of Reich (2013), therefore I am not surprised that the authors adopt the same terminology. Nevertheless, one should keep in mind that "particle rejuvenation" is not a new method invented by Reich and colleagues, it is just a new fancy name for one of the regularisation methods that have been introduced in the 2000s by Musso and colleagues. See, in particular, the chapter "Improving Regularized Particle Filters" by Musso et al. in the book "Sequential Monte Carlo Methods in Practice" by Doucet, Freitas, and Gordon (isbn: 978-0-387-95146-1). This historical perspective does not appear in the manuscript and I think that this is missing.

In addition, I would like to mention that in my opinion, "regularisation" is a better name than "particle rejuvenation", because I think that it better describes what is actually happening (in practice the posterior pdf is regularised). Of course, I acknowledge that the authors should have the right to choose which name they use!

**1.3 Particle filter and localisation**

The curse of dimensionality, mentioned in the introduction, is one of the main obstacles to the application of PF algorithms to high-dimensional problems (see Snyder et al., 2008, doi: 10.1175/2008MWR2529.1). More recently, a lot of studies have tried to apply localisation techniques in the PF to circumvent the curse of dimensionality (see in particular the review by Farchi and Bocquet, 2018, doi: 10.5194/npg-25-765-2018), which has lead to successful applications of PF algorithms to high-dimensional problems.

In the manuscript, there is no discussion about the scalability of the new method, at a point that a naive reader could think that this new method could be applied as is to high-dimensional problems. This aspect must be clarified in the manuscript. In particular, I think that the author should explain whether the new method should (i) replace localisation or (ii) be used in conjunction with localisation.

**1.4 Numerical experiments and conclusions**

After reading the article, I am left with the impression that the numerical experiments are very brief. Nowadays, a couple of experiments with the Lorenz 1963 model is not enough to publish an article in a data assimilation journal like NPG. Therefore, I think that Section 5 (with the numerical experiments) has to be extended. At the very least, I think that a test series with the Lorenz 1996 model should be included, which would help illustrate a potential discussion about the scalability of the new method. Of course, implementing a PF algorithm with 40 variables (the classical size of the Lorenz 1996 model) is a challenge without localisation, which is why it is probably more reasonable to start with only 10 or 8 variables in the Lorenz 1996 model.

In addition, the results of the numerical experiments are barely discussed in the manuscript, and the conclusions are very short (only two paragraphs!). I think that this

is clearly not enough and that the authors should provide appropriate discussion of the results and conclusions.

**2 Specific and technical comments**

**L. 12-13** "The curse of dimensionality". At this point, the authors could cite Snyder et al. (2008).

**L. 17** "by discarding information about the underlying dynamical system". In the EnKF, the information about the dynamical system is taken into account through the use of the dynamical model (for forecast) and the observation operator. Could the authors explain what they mean here?

**L. 18** "that lives in $\mathbb{R}^n$". At this point, $n$ is not defined. I would delay this aspect until section 2 where the different spaces are introduced.

**LL. 19-20** "into our assumed posterior normal distribution". If both the prior and the likelihood are assumed Gaussian, then the posterior is Gaussian (this is not an assumption).

**L. 21** In the reference list, two elements match the key "Popov et al., 2020". This should be corrected.

**L. 23** I would suggest the following stylistic transformation "(ETPF) (Reich, 2013)" $\rightarrow$ "(ETPF, Reich, 2013)".

**L. 25** "the ensemble limit". To my knowledge, this is not clearly defined in the data assimilation community (even though I agree that this is understandable). I would recommend to explicitly define this term with, for example: "in the limit of an infinite ensemble size".

**L. 26** "This means that". The logical connection is incorrect here.

**LL. 45-46** "and the supports of the probability densities $\pi_{X^f}$ and $\pi_{X^a}$ are subsets of the respective spaces". This could be reformulated because as is, one could understand that it is possible that the support of these pdfs are not subsets of the respective spaces.

**LL. 50 and 54** The sum's limits are incorrect in Eq. (2) and (3).

**L. 57** "ensemble of weights". I would suggest to name it "weight vector".

**L. 58** "Using (3) and (4) empirical estimates of the posterior mean and covariance". This seems weird. I would suggest a reformulation.

**L. 59** In equation (5), the authors use the prefactor $N/(N-1)$ to debias the sample covariances. However, for weighted sample the prefactor to debias the sample covariances is $1/(1 - \mathbf{w}^\top \mathbf{w})$, which is equal to $N/(N-1)$ only in the case where the weights are uniform $\mathbf{w} = \mathbf{1}_N/N$. Could the authors justify this choice?

**L. 61** "The goal of particle filtering (with resampling)". I would rather say that this is the goal of resampling.

**L. 62** "the the posterior..." $\rightarrow$ "the posterior..."

**LL. 65-66** "We impose that the empirical mean (5) is preserved by (6)". This is in general not possible with classical resampling algorithms. Of course, this is possible when using a linear ensemble transformation like Eq. (8) but at this point in the manuscript, Eq. (8) is not yet introduced!

**L. 71** "$\mathbf{T}^* \in \mathbb{R}^{N^f \times N^a}$" At this point, it could be interesting to remind the reader that $\mathbf{T}^*$ has positive coefficients.

**L. 73** "$\mathbf{T}^\top \mathbf{1}_{N^f} = \mathbf{1}_{N^f}$" $\rightarrow$ "$\mathbf{T}^\top \mathbf{1}_{N^f} = \mathbf{1}_{N^a}$"

**L. 78** "$X^a = \Psi(X^f)$, which has..." $\rightarrow$ "$X^a = \Psi(X^f)$ has..."

**L. 94** "the factor $\tau$". The authors could mention that $\tau$ is usually called the bandwidth.

**L. 99** The second line of Eq. (13) is just the transpose of the first line, or did I miss anything?

**L. 97-99** It is true that the extra term ensures that the regularisation noise has zero mean. However, the author should mention that this extra term does modify the sample covariance of the noise. The same holds for Eq. (26).

**L. 125** "THis" $\rightarrow$ "This"

**LL. 125-126** "This is because we are now incorporating more prior information $P$ in the form of climatological information". From what I understand, such additional information was missing until now. Hence Eq. (2) – and the few following equations – should be written "$\pi_{\hat{X}^f}(X)$" instead of "$\pi_{\hat{X}^f}(X|P)$", right?

**LL. 134-135**  "is assumed to be the sample mean of the dynamic ensemble" I would mention here that this choice is necessary to preserve the mean of the augmented ensemble.

**L. 136**  "by construction and (13), thus requiring that only the synthetic ensemble anomalies need to be determined". This is hardly understandable. Y would suggest a reformulation.

**L. 158**  Please define the "$\wedge$" symbol in Eq. (28).

**L. 162**  "Note that if $\mathcal{P} = \boldsymbol{\Sigma}_{X^{\mathrm{f}}}$...". How is the division by zero in Eq. (28) handled in this case?

**L. 162**  "In such a framework the scaling parameter...". I think that a separation is needed here to indicate that this does not apply to $\mathcal{P} = \boldsymbol{\Sigma}_{X^{\mathrm{f}}}$.

**LL. 170-171**  I understand that the scaling of $\mathcal{P}$ has no impact on the definition of $\boldsymbol{\Sigma}_{\mathcal{X}^{\mathrm{f}}}$ defined by Eq. (25), but does it have an impact on $\gamma$ defined by Eq. (28)?

**L. 175**  "where the optimal transport matrix $\mathbf{T}^* \in \mathbf{R}^{(N+M)\times N}$ is computed by solving (9)". This is a very concise description and I think that additional description is needed, because this is one of the core elements of the new method. At the very least, it should be mentioned that a posterior weight is needed for all members of the augmented ensemble, and that the formulation of (9) needs to be adjusted to take into account non-uniform prior weights. In addition, note that "$\mathbf{R}$" should be replaced by "$\mathbb{R}$".

**LL. 181-182**  "In effect we are able to avoid ensemble collapse by enhancing the empirical measure distribution (32) with new prior information". From a theoretical perspective, this has not been proven. This is only illustrated in Section 5 using numerical illustration with a 3-variable model. At this point, the lack of discussion about scalability really hurts (see general comment in section 1.3).

**LL. 227**  Why not use the time-averaged RMSE, defined as

$$\mathrm{RMSE}(\mathbf{x}^{\mathrm{t}}, \bar{\mathbf{x}}^{\mathrm{a}}) \triangleq \frac{1}{T}\sum_{i=1}^{T}\sqrt{\frac{1}{n}\sum_{j=1}^{n}\left([\mathbf{x}_i^{\mathrm{t}}]_j - [\bar{\mathbf{x}}_i^{\mathrm{a}}]_j\right)^2}, \tag{1}$$

which is the indicator used in most articles in data assimilation?

**LL. 230**  "with the optimal rejuvenation factor of $\tau = 0.04$". I highly doubt that $\tau = 0.04$ is optimal for all values of the ensemble size $N$. In order to make a fair comparison, the value of $\tau$ should be optimally tuned for each ensemble size $N$. If not, we give an unfair advantage to any of the method.

**LL. 235**  "Results in Figure 1 show...". I think that this test series (and the second one as well) is lacking a baseline. For this small 3-variable model, the baseline could be the score obtained with a classical PF (for example the SIR or Bootstrap filter) with a very large ensemble (typically more than $10^3$ particles) and with optimally tuned regularisation.

**LL. 241-243**  "For a low ensemble size... as compared to the ETPF". From what can be seen in Fig. 2, there is a difference between, for example, $\alpha = 1$ and $\alpha = 1.2$ (the latter being more accurate). It is possible that this difference falls into the variability between the 20 independent runs, but this is not clearly explained in the text or in the figure.

**LL. 259-260**  "We believe that the stochastic covariance shrinkage approach to importance sampling can be used not just for particle rejuvenation in the ETPF, but in other particle filters as well". Let us take the example of the most basic PF, the SIR filter. During the resampling step, some particles (typically those with low weight) are discarded and replaced by other particles (typically those with high weight). If applying the new proposed method, this would unavoidably lead to replacing original particles by synthetic particles, which is probably not something that is desirable. With this small example, I hope that I have convinced the authors that more discussion is needed here regarding the application of the new method to other PFs than the ETPF.

**Figs. 1 and 2**  Some of the lines cannot be distinguished when the manuscript is printed in black and white. This should be corrected. In addition, I would recommend a log scale for the x-axis and I would recommend to show the grid for clarity.

---

## Referee Comment (RC2)

Review of "A Stochastic Covariance Shrinkage Approach to Particle Rejuvenation in the Ensemble Transform Particle Filter"

by A. A. Popov et al.

In order to circunvent the collapse of the weights in the ensemble transform particle filter, this manuscript proposes to generate syntetic particles sampled from a normal distribution with the dynamical particle sample mean and a climatological covariance. Synthetic and dynamical particles are combined using weights from a covariance shrinkage estimator. The manuscript is in general well written and the proposed methodology may be an interesting venue for the particle filter limitations. However the methodology may require further fundamentation and a more extensive experimentation is required to evaluate the proposed "particle rejuvenation" in different regimes with a discussion of the limitations and strenghs of the method. An experiment in a higher dimensional system is also required. I consider this manuscript is suitable for publication in NGP however it requires a major revision.

**Some specific points:**

1. The discussion of convergence of the method is rather short (Ln 235). In principle, for a large number the dynamical particles the estimator will give weigth 0 to the synthetic particles. Is that correct? Is this the reason of the convergence? The authors should discuss how the estimator varies in the experiments as a function of the number of particles. In a regime with small/medium number of particles, there should be two effects that the methodology should lead to a suboptimal filter. The synthetic particles are sampled from a Gaussian distribution which may deteriorate the performance of the filter for non-Gaussian forecast predictions. In several dynamical systems the prediction covariance varies with time, then the use of prior information with a climatological covariance should give a suboptimal filter. The authors should discuss these limitations and evaluate them in the experiments.

2. A plot and/or a discussion in the experiments about the values given by the shrinkage estimator (which is used as weight in the dynamical and synthetic particles) are required.

3. To my understanding of the methodology, the authors are not using the sampled particles as a rejuvenation of the sequential filter but just to improve the inference step in the ETKF. The synthetic particles are not used in the prediction step, is this correct?. I had in mind that in the ETPF, the rejuvenated particles were used in the prediction step? Could the authors discuss this point in the manuscript?

4. Sampling perturbations from climatological covariances in geophysical applications may give physically unrealistic states. The authors should comment how the method could be extended to be applied in realistic applications.

5. Experiments are rather insufficient. An experiment with a nonlinear observational operator or any other configuration that result in a non-Gaussian posterior distribution would also be

illustrative. Experiments with different observational errors (particularly smaller ones) are also required.

6. The authors may compare the performance of the experiment with a standard inflation technique (with optimal inflation factor) as a baseline. They mentioned in passing that the ETKF deteriorates with inflation, a deeper examination of this point could be useful to further motivate the proposed method.

7. The experiments comparing the Laplacian with Gaussian sampling, and the ones comparing two climatological covariances versus one do not appear to have a conclusion.

8. I had in mind covariance shrinkage as a covariance regularization method for small samples. However, the low-dimensional example shown in the manuscript does not appear to evaluate the regularization of the long-distance correlations. The authors should evaluate at least in a 40-dimension Lorenz-96 the performance of the methodology that they are proposing. In principle one expects a larger impact of the covariance shrinkage estimator in higher dimensional systems.

---

## Author Comment (AC1)

**Response to Referees**

Andrey A Popov et al.

January 31, 2022

**1 Misc. Changes**

Three figures have been added. One, illustrating the optimal transport methodology in the continuous one dimensional case. We hope that this figure would help the reader visually understand the optimal transport methodology. One illustrating the Stochastic shrinkage approach. One illustrating R-Localization.

**2 Response to Referee 1**

**2.1 General Comments**

**C** Throughout the manuscript, the notation is inconsistent. For example, in Eq. (2) the argument if X|P while in Eq. (3) it is x|y, p. Using a consistent notation would really make the manuscript clearer and hence help the reader. Furthermore, I strongly recommend to follow the usual conventions of the data assimilation community (which, if I am not mistaken, also coincide with the journal conventions): • bold face uppercase for matrices (ex M); • bold face lowercase italic for vectors (ex v); • lowercase italic or greek letters for scalar quantities (ex n or ...); 1 • uppercase italic for sizes (ex N).

We have changed as much notation as possible without interrupting the flow of what we feel the text represents. We have left some of the more novel notation alone (such as the use of caligraphy and fraktur) as it is in line with previous work, which we strongly believe is important for consistency.

1.2 Regularisation or particle rejuvenation The entire method derived by the authors is designed as a sort of extension of the ETPF of Reich (2013), therefore I am not surprised that the authors adopt the same terminology. Nevertheless, one should keep in mind that "particle rejuvenation" is not a new method invented by Reich and colleagues, it is just a new fancy name for one of the regularisation methods that have been introduced in the 2000s by Musso and colleagues. See, in particular, the chapter "Improving Regularized Particle Filters" by Musso et al. in the book "Sequential Monte Carlo Methods in Practice" by Doucet, Freitas, and Gordon (isbn: 978-0-387-95146-1). This historical perspective does not appear in the manuscript and I think that this is missing. In addition, I would like to mention that in my opinion, "regularisation" is a better name than "particle rejuvenation", because I think that it better describes what is actually happening (in practice the posterior pdf is regularised). Of course, I acknowledge that the authors should have the right to choose which name they use!

We greatly appreciate the historical context given by the reviewer. While we did not make the connection in the manuscript before, we are aware of the strong connections between regularization and rejuvenation. We however believe that they are not one and the same. From our point of view regularization is an action on the probability distribution, while rejuvenation is an action on the samples. We added two lines about this in the introduction of the paper.

1.3 Particle filter and localisation The curse of dimensionality, mentioned in the introduction, is one of the main obstacles to the application of PF algorithms to high-dimensional problems (see Snyder et al., 2008, doi: 10.1175/2008MWR2529.1). More recently, a lot of studies have tried to apply localisation techniques in the PF to circumvent the curse of dimensionality (see in particular the review by Farchi and Bocquet, 2018, doi: 10.5194/npg-25-765-2018), which has lead to successful applications of PF algorithms to high-dimensional problems. In the manuscript, there is no discussion about the scalability of the new method, at a point that a naive reader could think that this new method could be applied as is to high-dimensional problems. This aspect must be clarified in the manuscript. In particular, I think that the author should explain whether the new method should (i) replace localisation or (ii) be used in conjunction with localisation.

We have cited both the papers provided, and have included a short discussion of high-dim particle filters in the introduction.

フフ

1.4 Numerical experiments and conclusions After reading the article, I am left with the impression that the numerical experiments are very brief. Nowadays, a couple of experiments with the Lorenz 1963 model is not enough to publish an article in a data assimilation journal like NPG. Therefore, I think that Section 5 (with the numerical experiments) has to be extended. At the very least, I think that a test series with the Lorenz 1996 model should be included, which would help illustrate a potential discussion about the scalability of the new method. Of course, implementing a PF algorithm with 40 variables (the classical size of the Lorenz 1996 model) is a challenge without localisation, which is why it is probably more reasonable to start with only 10 or 8 variables in the Lorenz 1996 model. In addition, the results of the numerical experiments are barely discussed in the manuscript, and the conclusions are very short (only two paragraphs!). I think that this is clearly not enough and that the authors should provide appropriate discussion of the results and conclusions.

In line with this notion and the next, we have added a localized FETPF, which we name the LFETPF. We have added experiments with the Lorenz '96 equations, and have shown that the new methodology can produce good results with as few as 4 dynamical ensemble members.

We are very grateful to the reviewer for pushing us in this direction, as we believe this experiment makes the results provided in the manuscript stronger.

**3** Technical Corrections**

**C** L. 12-13 "The curse of dimensionality". At this point, the authors could cite Snyder et al. (2008).

77

77

"

77

We agree and have cited the paper.

**C** L. 17 "by discarding information about the underlying dynamical system". In the EnKF, the information about the dynamical system is taken into account through the use of the dynamical model (for forecast) and the observation operator. Could the authors explain what they mean here?

This is a largely philosophical point, which we believe that is worth making. Most dynamical systems of interest have compact support on complex and hard-to-describe manifolds. This is the "information" that the agent performing the inference has in possession, and must use for the inference to be Bayesian. This "information" is discarded when the ensemble of realizations is assumed Gaussian. Thus, the information about the dynamical system is only partially taken into account. We have modified this line to say "by partially discarding".

**C** L. 18 "that lives in Rn". At this point, n is not defined. I would delay this aspect until section 2 where the different spaces are introduced.

We have removed the reference to n and describe it as the "state space".

**C** LL. 19-20 "into our assumed posterior normal distribution". If both the prior and the likelihood are assumed Gaussian, then the posterior is Gaussian (this is not an assumption).

This is again a matter of philosophy. As we are assuming a Gaussian prior, and we know that in actuality the distribution is not and cannot be Gaussian, we know that we are not performing Bayesian inference. Thus, Bayes' theorem no longer applies to what we are doing. We therefore have to make assumptions about the posterior.

We recognize the point made by the reviewer, and we understand that his point of view would be more common in the community. However, we would like to be technically correct rather than conventionally correct.

L. 21 In the reference list, two elements match the key "Popov et al., 2020". This should be corrected.

In our understanding this is an issue with the Copernicus bibliography style. As we have no way of fixing this without modifying the provided style, we do not have a way of modifying this.

**L**. 23 I would suggest the following stylistic transformation "(ETPF) (Reich, 2013)"  $\rightarrow$  "(ETPF, Reich, 2013)".

We agree with the reviewer and have fixed it as best as is possible within the Copernicus style convention.

L. 25 "the ensemble limit". To my knowledge, this is not clearly defined in the data assimilation community (even though I agree that this is understandable). I would recommend to explicitly define this term with, for example: "in the limit of an infinite ensemble size". We agree with the reviewer and have made this change. " L. 26 "This means that". The logical connection is incorrect here. We changed the wording to add the modal "possible". **LL**. 45-46 "and the supports of the probability densities  $\pi(Xf)$  and  $\pi(Xa)$  are subsets of the respective spaces". This could be reformulated because as is, one could understand that it is possible that the support of these pdfs are not subsets of the " respective spaces. We have reworded this in a clearer way. " LL. 50 and 54 The sum's limits are incorrect in Eq. (2) and (3). We have corrected this. L. 57 "ensemble of weights". I would suggest to name it "weight vector". " While we agree that notationally this should be a vector, we prefer using ensemble of weights rather than weight vector to make the link more obvious between the ensemble and the corresponding weights of each ensemble member. " L. 58 "Using (3) and (4) empirical estimates of the posterior mean and covariance". " This seems weird. I would suggest a reformulation. We have reformulated this. " L. 59 In equation (5), the authors use the prefactor N/(N - 1) to debias the sam-

L. 59 In equation (5), the authors use the prefactor N/(N - 1) to debias the sample covariances. However, for weighted sample the prefactor to debias the sample covariances is 1/(1 - ww), which is equal to N/(N - 1) only in the case where the weights are uniform w = 1N/N. Could the authors justify this choice?

This formula has been corrected. As we do not use this formula in the implementation, this had no consequences on the results.

"

L. 61 "The goal of particle filtering (with resampling)". I would rather say that this is the goal of resampling.

77

Since what we are referring to is the goal of particle filtering and also resampling (which is one solution to particle filtering), we have reworded this as "Our goal".

L. 62 "the the posterior..."  $\rightarrow$  "the posterior..."

We have made this correction.

LL. 65-66 "We impose that the empirical mean (5) is preserved by (6)". This is in general not possible with classical resampling algorithms. Of course, this is possible when using a linear ensemble transformation like Eq. (8) but at this point in the manuscript, Eq. (8) is not yet introduced!

"

"

"

"

77

"

This has been changed to "our goal" in the begining, thus circumventing the need for a discussion about resampling.

L. 71 "..." At this point, it could be interesting to remind the reader that T has positive coefficients.

Agreed. We have done this change.

**L**. 73 L. 78

"

"

"

"

"

"

Both these formulae have been changed.

**66** L. 94 "the factor  $\tau$ ". The authors could mention that  $\tau$  is usually called the bandwidth.

We have added this.

L. 99 The second line of Eq. (13) is just the transpose of the first line, or did I miss anything?

It is indeed the case. We include it for the sake of completeness. We believe that it does not leave any guesswork for the reader.

L. 97-99 It is true that the extra term ensures that the regularisation noise has zero mean. However, the author should mention that this extra term does modify the sample covariance of the noise. The same holds for Eq. (26).

Taking out the sample mean does not change the sample covariance. We are not sure what the reviewer means by this.

L. 125 "THis"  $\rightarrow$  "This"

This has been corrected.

LL. 125-126 "This is because we are now incorporating more prior information P in the form of climatological information". From what I understand, such additional information was missing until now. Hence Eq. (2) – and the few following equations – should be written " $\pi$  Xf(X)" instead of " $\pi$  Xf(X|P)", right?

"

"

This "additional information" is always present. Bayesian inference needs to take into account all information. Most of the time in practice, the vast majority of information is ignored, (meaning that almost all practical inference is in fact non-Bayesian) but that does not mean it is not there in the ideal case. Instead of ignoring this information, we recognize that it is always there. We refer the reviewer to the book "Probability theory: The logic of science" by Jaynes for more information.

**C** LL. 134-135 "is assumed to be the sample mean of the dynamic ensemble" I would mention here that this choice is necessary to preserve the mean of the augmented ensemble.

We agree and have fixed this.

"

**C** L. 136 "by construction and (13), thus requiring that only the synthetic ensemble anomalies need to be determined". This is hardly understandable. I would suggest a reformulation.

We have reformulated this.

L. 158 Please define the  $\land$  symbol in Eq. (28).

We have used min instead of the AND operation.

L. 162 "Note that if P = ...". How is the division by zero in Eq. (28) handled in this case?

We have clarified this.

L. 162 "In such a framework the scaling parameter...". I think that a separation is needed here to indicate that this does not apply to P = ....

A paragraph separation was added.

**C** LL. 170-171 I understand that the scaling of P has no impact on the definition of ... defined by Eq. (25), but does it have an impact on  $\gamma$  defined by Eq. (28)?

We have expanded the remark to address that  $\gamma$  is not scaled by scaling P.

L. 175 "where the optimal transport matrix ... is computed by solving (9)". This is a very concise description and I think that additional description is needed, because this is one of the core elements of the new method. At the very least, it should be mentioned that a posterior weight is needed for all members of the augmented ensemble, and that the formulation of (9) needs to be adjusted to take into account non-uniform prior weights. In addition, note that "R" should be replaced by "R".

This paragraph has been expanded to mention the prior information weights and the importance sampling procedure.

**C** LL. 181-182 "In effect we are able to avoid ensemble collapse by enhancing the empirical measure distribution (32) with new prior information". From a theoretical perspective, this has not been proven. This is only illustrated in Section 5 using numerical illustration with a 3-variable model. At this point, the lack of discussion about scalability really hurts (see general comment in section 1.3).

We have changed this statement to be much weaker. "We attempt to avoid".

LL. 227 Why not use the time-averaged RMSE, defined as ...

We looked at the four major books on data assimilation:

- Evensen (2009) uses spatio-temporal RMSE
- Reich and Cotter equation (1.13) is also spatio-temporal RMSE (it is mis-named in the text, but it is spatio-temporal). See also example 8.9 again given the spatio-temporal RMSE equation.
- Law Stuart and Zygalakis uses time-averaged RMSE
- · Asch, Bocquet and Nodet use time-averaged RMSE

We hope the reviewer agrees that it seems like there is no clear consensus in the literature. We prefer the spatio-temporal RMSE.

**C** LL. 230 "with the optimal rejuvenation factor of  $\tau = 0.04$ ". I highly doubt that  $\tau = 0.04$  is optimal for all values of the ensemble size N. In order to make a fair comparison, the value of  $\tau$  should be optimally tuned for each ensemble size N. If not, we give an unfair advantage to any of the method.

We have removed the "optimal" word and instead cited the ETPF paper where the author used the same value for all experiments. Note that since our rejuvenation is a modification of the cited paper, the parameter is a square of that used in the cited paper.

"

**C** LL. 235 "Results in Figure 1 show...". I think that this test series (and the second one as well) is lacking a baseline. For this small 3-variable model, the baseline could be the score obtained with a classical PF (for example the SIR or Bootstrap filter) with a very large ensemble (typically more than 103 particles) and with optimally tuned regularisation.

We have added SIR baselines to the first two experiments.

**C** LL. 241-243 "For a low ensemble size... as compared to the ETPF". From what can be seen in Fig. 2, there is a difference between, for example,  $\alpha = 1$  and  $\alpha = 1.2$  (the latter being more accurate). It is possible that this difference falls into the variability between the 20 independent runs, but this is not clearly explained in the text or in the figure.

Both points are outside of 3 standard deviations of each other. We have added a comment that all reported differences are for values more than 3 standard deviations apart.

LL. 259-260 "We believe that the stochastic covariance shrinkage approach to importance sampling can be used not just for particle rejuvenation in the ETPF, but in other particle filters as well". Let us take the example of the most basic PF, the SIR filter. During the resampling step, some particles (typically those with low weight) are discarded and replaced by other particles (typically those with high weight). If applying the new proposed method, this would unavoidably lead to replacing original particles by synthetic particles, which is probably not something that is desirable. With this small example, I hope that I have convinced the authors that more discussion is needed here regarding the application of the new method to other PFs than the ETPF.

We agree and this piece of text has been removed.

Figs. 1 and 2 Some of the lines cannot be distinguished when the manuscript is printed in black and white. This should be corrected. In addition, I would recommend a log scale for the x-axis and I would recommend to show the grid for clarity.

We have remade all the figures with these suggestions in mind.

**4 **Response to Referee 2**

1. The discussion of convergence of the method is rather short (Ln 235). In principle, for a large number the dynamical particles the estimator will give weight 0 to the synthetic particles. Is that correct? Is this the reason of the convergence? The authors should discuss how the estimator varies in the experiments as a function of the number of particles. In a regime with small/medium number of particles, there should be two effects that the methodology should lead to a sub-optimal filter. The synthetic particles are sampled from a Gaussian distribution which may deteriorate the performance of the filter for non-Gaussian forecast predictions. In several dynamical systems the prediction covariance varies with time, then the use of prior information with a climatological covariance should give a sub-optimal filter. The authors should discuss these limitations and evaluate them in the experiments.

"

We have added a section on convergence, and showed that the FETPF converges in the limit of dynamical ensemble size (if some conditions are met).

**6** 2. A plot and/or a discussion in the experiments about the values given by the shrinkage estimator (which is used as weight in the dynamical and synthetic particles) are required.

"

We have added histograms of the distributions in a new plot.

**3**. To my understanding of the methodology, the authors are not using the sampled particles as a rejuvenation of the sequential filter but just to improve the inference step in the ETKF. The synthetic particles are not used in the prediction step, is this correct?. I had in mind that in the ETPF, the rejuvenated particles were used in the prediction step? Could the authors discuss this point in the manuscript?

Rejuvenation for the ETPF refers to a stochastic regularization by adding random perturbations to the existing ensemble after the inference step. Purely random data is not predicted upon. In the FETPF the only difference is the synthetic data is combined during the inference step instead of after, in a more informed manner. We have added a discussion of regularization that hopefully clarifies this a bit more in the text.

4. Sampling perturbations from climatological covariances in geophysical applications may give physically unrealistic states. The authors should comment how the method could be extended to be applied in realistic applications.

We have added a remark about physically unrealistic realization in the convergence section. In both the ETPF and FETPF, in the limit of ensemble size, physically realistic realization are generated with probability one. This is now explicitly stated.

5. Experiments are rather insufficient. An experiment with a nonlinear observational operator or any other configuration that result in a non-Gaussian posterior distribution would also be illustrative. Experiments with different observational errors (particularly smaller ones) are also required.

We have significantly expanded on the experiments. We have added two experiments with Lorenz '96 with linear and non-linear observations.

As for small observation error: if we have a small Gaussian observation error, then the analysis distribution would tend towards being Gaussian. As we wish to examine the case of a non-Gaussian analysis distribution, a large observation error is chosen.

6. The authors may compare the performance of the experiment with a standard inflation technique (with optimal inflation factor) as a baseline. They mentioned in passing that the ETKF deteriorates with inflation, a deeper examination of this point could be useful to further motivate the proposed method.

We are not sure as to what the reviewer is referring. No inflation is performed on the dynamical ensemble anywhere in the paper. Additionally, the Lorenz '63 cases discussed are sufficiently non-Gaussian such that no ensemble Kalman-based filter ever converges for the given problem.

77

New experiments with the Lorenz '96 equations do look at the ETKF, and thus approximately optimal inflation factors are used.

**5.** 7. The experiments comparing the Laplacian with Gaussian sampling, and the ones comparing two climatological covariances versus one do not appear to have a conclusion.

All conclusions have been rewritten and the experiments part significantly expanded. This is hopefully not an issue anymore.

**8**. I had in mind covariance shrinkage as a covariance regularization method for small samples. However, the low-dimensional example shown in the manuscript does not appear to evaluate the regularization of the long-distance correlations. The authors should evaluate at least in a 40-dimension Lorenz-96 the performance of the methodology that they are proposing.In principle one expects a larger impact of the covariance shrinkage estimator in higher dimensional systems.

40-variable localized Lorenz '96 experiments have been added, with both linear and nonlinear observation operators. We hope that these new experiments satisfy all concerns.

"

---

## Referee Report (RR1)

**A stochastic covariance shrinkage approach to particle rejuvenation in the ensemble transform particle filter – Second review report**

25th February 2022

In the revised manuscript, the authors have corrected the manuscript to take into account the remarks of both reviewers. In particular they have added a couple methodological sections about localisation and convergence, as well as a new set of experiments with the 40-variable Lorenz 1996 model. The additional content overall contributes to the improvement of the manuscript. However, I also have the impression that the authors did not address all the reviewers' concerns.

**1 General comments**

**1.1 Lorenz 1963 test series**

This first test series with the Lorenz 1963 model is using the exact same setup as one of the experiments described by Acevedo et al. (2017), hereafter A17. However, the results reported in the present manuscript (Figs. 4 and 5) are different from the ones reported by A17 (left panel of Fig. 7.1). In particular, I noticed the following differences in score:

- SIR benchmark: 2 here vs. 1.5 in A17 (dark red line);

- EnKF: diverged here vs. 2.5 in A17 (cyan line);

- ETPF with 15 particles: 12 here vs. 7.5 in A17;

- ETPF with 25 particles: 6 here vs. 4 in A17;

- ETPF2 with 15 particles: 5 here vs. 3.5 in A17;

- ETPF2 with 25 particles: 3.5 here vs. 2.5 in A17.

How can such differences be explained?

In addition, I would like to come back on the choice of the rejuvenation factor. In general, the optimal rejuvenation factor depends on the ensemble size. For a small range of values of the ensemble size, such as [15, 35] as used by A17, using a constant factor for all values of the ensemble size may be a good approximation. For a larger range of values of the ensemble size, such as [5, 100] as used in the present manuscript, this approximation is less justified. Therefore, I think that the present test series should include a tuning of the rejuvenation factor which depends on the ensemble size, or at least test different values as done by A17.

Finally, I would like to mention that I appreciate the efforts that the authors have put into the improvement of their figures. I have one last remark: the dashed lines can not be distinguished from the plain lines in the legend of Figs. 4, 5, 8, and 9.

**1.2 Lorenz 1996 test series**

With the linear observation operator, this test series uses a standard and well-documented setup, which is commonly used to assess the performance of new data assimilation algorithms. Once again, the results some results are different from what can be found in the literature:

- the LETKF curve in Fig. 8 does not seem to be correct: the RMSE should be lower than 0.3 with 5 members, close to 0.2 with 10 members, and lower than 0.2 (these scores can be found, for example, in the chapter on the EnKF of Ash et al., 2016, already cited in the manuscript);

- the authors mention that the LETPF does not converge, but Farchi and Bocquet (2018) provide an illustration of the convergence of the LETPF in the exact same setup (red curve in Fig. 16 of their article), with lower RMSE scores as those reported in Fig. 8 of the present manuscript for the LFETPF(G) with 8 and 16 particles.

My intuition is that these differences can be largely explained by the (very restrictive) choice of not tuning the localisation radius and the inflation or rejuvenation factor.

With the non-linear observation operator, the author conclude that the setup is "highly Gaussian". My impression while reading the text is that the purpose of this setup is precisely to be non-Gaussian. If we end up with a Gaussian setup, then this does not provide any added value compared to the first setup. In addition, there is a contradiction with the conclusion of the experiments with the linear observation operator: in a highly Gaussian setup the LETKF outperforms the LFETPF (which is expected and which is not what can be seen in the experiments with the non-linear observation operator).

**1.3 General comments on the response to referee 1**

In many cases, the corrections described in the answers do not match the revised manuscript (answer to comments L. 17, L. 18, L. 26, LL. 45-46, and L.58). A mistake

can happen, but at this point this is unprofessional and a clear waste of time for both authors and reviewers.

**2 Specific and technical comments on the revised manuscript**

**LL. 17-19**  "Recent attempts to apply particle filters [...] methods such as the ensemble Kalman filter." I appreciate this additional discussion about localisation in the particle filter which was needed. However, having this discussion right here seems weird. Indeed, at this point, the particle filter has not been introduced, neither has the weight collapse phenomenon. For this reason, I would suggest to move this discussion to the end of the following paragraph (namely after "Like all particle filters, ETPF is susceptible to weight collapse"). Also please correct the citation to Farchi and Bocquet, 2018.

**L. 22**  "ETPF transports..." (and many other occurences) "ETPF" is an acronym and not a name, so I would suggest to use "the ETPF" instead of just "ETPF".

**L. 42**  With the new notation, I think that one should read "$\pi_{Y|X^{\mathrm{f}}}$" instead of "$\pi_{Y|X}$" in Eq. (1).

**L. 54**  Please define "$\hat{X}^{\mathrm{a}}$" right after Eq. (3) and not later (at the moment it is defined L.65).

**L. 102**  "Specifically, consider the ensembles the $l^{th}$ state space variables:" I think that this formulation is incorrect.

**L. 107**  "$\mathbf{T}^{\top}\mathbf{1}_{N^{\mathrm{f}}} = \mathbf{1}_{N^{\mathrm{f}}}$" → "$\mathbf{T}^{\top}\mathbf{1}_{N^{\mathrm{f}}} = \mathbf{1}_{N^{\mathrm{a}}}$". This is the same mistake as in the original manuscript.

**L. 187**  "form" → "from".

**L. 213**  "In effect attempt to avoid ensemble collapse" I think that this formulation is incorrect.

**LL. 307-308**  "As the covariance chosen depends on the dynamical ensemble, these results indicate that a more detailed climatological distribution that varies seasonally might induce an even greater decrease in error." I do not understand the meaning of this sentence. Could it be clarified?

**L. 317**  "RMSEfor" → "RMSE for".

**L. 332**  "significantly much more significantly skewed" Please reformulate.

**LL. 353-354**   "[...] is not shown due to space limitations" What kind of limitations does prevent the authors from showing $\mathcal{P}$, which is a $40 \times 40$ matrix? If they want to, they could easily add a figure showing $\mathcal{P}$, for example in a similar way as Fig. 6.

**3 Specific and technical comments on the response to referee 1**

**Answer to comment 1.4**   In this comment, I mentioned that the discussion on the results and the conclusions are too short. This part of the comment has not been answered. The revised manuscript offers some discussion of the results, but in my opinion this is still not enough. Furthermore, the conclusions of the revised manuscript are even shorter than the original one!

**Answer to comment L. 97-99**   Clearly my remark was incorrect. I would like to apologise for this.

**Answer to comment LL. 125-126**   Visibly my remark has been misunderstood. In my opinion, there are two possible ways of rigorously treating the additional information:

1. either $P$ in Eqs. (1)-(6) does not include the climatological information but $P$ in Eq. (21) does include this climatological information, in which case it should be clearly mentioned that $P$ does not have the same meaning in Eqs. (1)-(6) and in Eq. (21);

2. or $P$ in Eqs. (1)-(6) does include the climatological information, but then Eqs. (1)-(6) are incorrect because, as mentioned in the text, this climatological information is ignored before Section 4: the conditional on $P$ should be removed.

**Answer to comment LL. 175**   Once again, I repeat that "$\mathbf{R}$" should be replaced by "$\mathbb{R}$".

---

## Editor Decision (ED1)

Dear Colleague,

Two referees have submitted their reviews of the revised version of your paper. They are the same referees as those of the first version, with the same identification numbers. In particular, referee 1, who had already let his name known, is Alban Farchi, from CEREA École des Ponts ParisTech in France.

As concerns Referee 2, he/she writes *The authors have addressed all of my comments. The manuscript has improved substantially with the added figures.* He/she recommends acceptance of the paper as it is.

A. Farchi is more critical, and makes a number of comments and suggestions, of actually unequal importance. He mentions that some of your results are in contradiction with results obtained by other authors in similar circumstances. This must certainly be mentioned in your paper. But it may be very difficult, if not impossible, to explain those contradictions. That may actually require additional experiments, either by you or the concerned authors. So do mention those contradictions and discuss them in the light of what you have done, as well as what you can tell concerning the other authors.

One of A. Farchi's comments is that he finds that your conclusion is too succinct. It is certainly succinct and, if you think you can say more (for instance, as concerns the possible limitations of your approach, or as to which difficulties could be expected in large dimension systems), please do so.

A. Farchi makes a number of editing comments. I as editor have also comments. There are actually a fairly large number of inconsistencies of notations, as well as typos. In addition to those that are mentioned by A. Farchi, I have noticed the following ones

1. L. 44, … *conditioned by the forecast* … I understand the forecast is $X^f$. Say it explicitly.

2. I do not see the usefulness of putting indices to the symbol $\pi$ when the latter designates a probability distribution. Those indices do not seem to bring any information that is not contained in the first of the arguments of $\pi$. Not only that is useless, but it even brings confusion. What is the meaning of the index $Y|X$ in $\pi_{Y|X}$ in Eq. (1) ? Why not simply $\pi_Y$ (see also the lines 44-45 that follow) ?

3. Why use $X^f$ in Eq. (1) and $\char"005E X^f$ in Eq. (2) ? What is the difference, if any ? (same remark concerning $X^a$ in Eqs 1 and 3).

4. Eq. (4)   $\pi_P (P | X^f_j) \rightarrow \pi_X (X^f_j | P)$

5. Eq. (5). Subscript $j$ on rhs is useless (and confusing) (incidentally, I do not understand the presence of the denominator $1 - w^{a,\mathrm{T}} w^{a,}$ in that equation. A brief explanation could be useful).

6. Eq. 7). Second sum runs from $j=1$ to $N^a$.

7. Eq. (9). $\mathbf{X}^f_k \rightarrow \mathbf{X}^a_k$ (see also Eq. 14)

8. Eq. (11), argument of the exponential. $\mathbf{R} \rightarrow \mathbf{R}^{-1}$

9. Eq. (12). Symbol o not defined (defined only on occasion of Eq. 46)

10. L. 357, *lead → led*

The list above is certainly not exhaustive. Please revise your paper, taking all A. Farchi's comments and suggestions, as well as my own, into account. Check in particular carefully and systematically all equations in the paper. And (except for minor typing corrections) answer precisely to all of our comments and suggestions. Should you disagree with a particular comment, or decide not to follow a particular suggestion, please state precisely your reasons for that.

I look forward to receiving a new version of your paper. I may send it to A. Farchi for an additional advice.

---

## Author Response (AR2)

**Response to Referees**

Andrey A Popov et al.

April 27, 2022

**1   General Changes**

Another figure has been added to showcase the Lorenz '96 covariance matrix.
    The notation concerning probability distributions has been largely simplified.

**2   Response to the Editor**

> ... He mentions that some of your results are in contradiction with results obtained by other authors in similar circumstances. This must certainly be mentioned in your paper. But it may be very difficult, if not impossible, to explain those contradictions.

We believe we have a simple explanation, namely the type of RMSE metric that is used. See the reply to his comments below. We believe that there is in fact no contradiction between our work and the paper cited in the review, and the results are consistent if the same RMSE formulas are used.

> he finds that your conclusion is too succinct. It is certainly succinct and, if you think you can say more (for instance, as concerns the possible limitations of your approach, or as to which difficulties could be expected in large dimension systems), please do so.

We have significantly expanded the conclusions to include all possible limitations and some insights as to what could be done in the future to further the research into this method and the ETPF in general.

> 2. I do not see the usefulness of putting indices to the symbol $\pi$ when the latter designates a probability distribution.

We agree and have simplified the notation greatly.

> 3. Why use $X^f$ in Eq. (1) and $\hat{X}^f$ in Eq. (2) ? What is the difference, if any ? (same remark concerning Xa in Eqs 1 and 3)

We have removed $\hat{X}^f$ and kept $\hat{X}^a$. We added a sentence to say that $\hat{X}^a$ converges in distribution to $X^a$. It exists to highlight the fact that the distribution for a finite ensemble is not exact, and that with perfect inference between the two empirical distributions we will not get perfect inference between the underlying random variables.

> 4. Eq. (4) $\pi(P|X) \rightarrow \pi(X|P)$
>
> 5. Eq. (5). Subscript j on rhs is useless (and confusing) (incidentally, I do not understand the presence of the denominator 1 – wa, T wa, in that equation. A brief explanation could be useful).

Both points have been corrected and carefully explained.

> 6. Eq. 7). Second sum runs from j=1 to $N^a$.

We fixed this value to $N^f$, as the ensemble of analysis weights is actually applied to the forecast ensemble, thus should have that size. It is a bit counter-intuitive, but this is how all particle filters operate.

> 7. Eq. (9). $X^f \rightarrow X^a$ (see also Eq. 14)

This is in fact $X^f$, as the transport is defined on the forecast ensemble distances.

> 8. Eq. (11), argument of the exponential. $R \rightarrow R^{-1}$ 9. Eq. (12). Symbol $\circ$ not defined (defined only on occasion of Eq. 46) 10. L. 357, lead $\rightarrow$ led

We agree with all of these, and have made the necessary changes.

**3  Response to Referee 1**

**3.1  Lorenz 63 Test series**

> This first test series with the Lorenz 1963 model is using the exact same setup as one of the experiments described by Acevedo et al. (2017), hereafter A17. However, the results reported in the present manuscript (Figs. 4 and 5) are different from the ones reported by A17 (left panel of Fig. 7.1). In particular, I noticed the following differences in score...

We have conducted experiments to determine why there are discrepancies, and came to the conclusion that this is again a matter of metric used. We use the spatio-temporal RMSE

$$\sqrt{\frac{1}{nT} \sum_{\text{time}} \sum_{\text{space}} \text{err}^2}, \tag{1}$$

while A17 uses the time-averaged RMSE

$$\frac{1}{T} \sum_{\text{time}} \sqrt{\frac{1}{n} \sum_{\text{space}} \text{err}^2}, \tag{2}$$

as evidenced by "the resulting time-averaged RMS errors" note in their manuscript. We believe that this is strong motivation for us to include the exact metric we used as a separate identifiable equation so that our results are more reproducible.

As we have shown the reviewer previously, there is no wide consensus between which version of the RMSE to use. Indeed the last author on A17 himself has used the other RMSE metric in other works, further confusing the matter.

In short, we do not believe there are any discrepancies between A17 and our work.

> In addition, I would like to come back on the choice of the rejuvenation factor. In general, the optimal rejuvenation factor depends on the ensemble size. For a small range of values of the ensemble size, such as [15, 35] as used by A17, using a constant factor for all values of the ensemble size may be a good approximation. For a larger range of values of the ensemble size, such as [5, 100] as used in the present manuscript, this approximation is less justified. Therefore, I think that the present test series should include a tuning of the rejuvenation factor which depends on the ensemble size, or at least test different values as done by A17.

We hope the reviewer agrees that the factor that we use should be close to optimal for some range of ensemble size between 5 and 100. As the ETPF performs worse for the whole range, we do not believe further exploration of this is warranted.

> Finally, I would like to mention that I appreciate the efforts that the authors have put into the improvement of their figures. I have one last remark: the dashed lines can not be distinguished from the plain lines in the legend of Figs. 4, 5, 8, and 9.

We have modified the lines in the legends to be a bit longer, thus making the distinction more clear.

**3.2   Lorenz 1996 test series**

> the LETKF curve in Fig. 8 does not seem to be correct: the RMSE should be lower than 0.3 with 5 members, close to 0.2 with 10 members, and lower than 0.2 (these scores can be found, for example, in the chapter on the EnKF of Ash et al., 2016, already cited in the manuscript);

As in the above, this difference is largely due to our use of a different RMSE measure coupled with all the other differences. As additional note of difference is with our GC implementation. We use the internal parameter $\theta$ that is not equal to one to more closely match Gaussian localization. This has been added to the text.

> the authors mention that the LETPF does not converge, but Farchi and Bocquet (2018) provide an illustration of the convergence of the LETPF in the exact same setup (red curve in Fig. 16 of their article), with lower RMSE scores as those reported in Fig. 8 of the present manuscript for the LFETPF(G) with 8 and 16 particles.
>
> ...
>
> My intuition is that these differences can be largely explained by the (very restrictive) choice of not tuning the localisation radius and the inflation or rejuvenation factor.

We agree with the reviewer on this point. The current study does not aim to optimally tune these factors through an exhaustive parameter search, however, we have taken a close look at the insight in the referenced work, and have done our best to choose parameters that give the LETPF the greatest chance of convergence through a lot of careful hand-tuning. As a result, the experiments have been modified in the text with a smaller localization radius.

> With the non-linear observation operator, the author conclude that the setup is "highly Gaussian". My impression while reading the text is that the purpose of this setup is precisely to be non-Gaussian. If we end up with a Gaussian setup, then this does not provide any added value compared to the first setup. In addition, there is a contradiction with the conclusion of the experiments with the linear observation operator: in a highly Gaussian setup the LETKF outperforms the LFETPF (which is expected and which is not what can be seen in the experiments with the non-linear observation operator).

We agree with the reviewer on this fact. This is purely an error on our part. We have completely revamped the nonlinear localization experiments to account for the facts presented therein.

**3.3  Specific Comments**

All minor comments have been addressed we hope this time.

> What kind of limitations does prevent the authors from showing P, which is a 40 × 40 matrix? If they want to, they could easily add a figure showing P, for example in a similar way as Fig. 6.

A figure showing the L96 $\mathcal{P}$ has been added.

**3.4  Technical Comments**

> In this comment, I mentioned that the discussion on the results and the conclusions are too short. This part of the comment has not been answered. The revised manuscript offers some discussion of the results, but in my opinion this is still not enough. Furthermore, the conclusions of the revised manuscript are even shorter than the original one!

We have expanded the conclusions to include many limitations of the work, and some more discussion on future possibilities. We have also expanded the discussion of the Lorenz '96 approach.

> However, having this discussion right here seems weird. Indeed, at this point, the particle filter has not been introduced, neither has the weight collapse phenomenon. For this reason, I would suggest to move this discussion to the end of the following paragraph

This has been changed as suggested.

> Also please correct the citation to Farchi and Bocquet, 2018.

The bib entry was auto-generated by Google. It seems to have been corrected there. We have corrected it in the manuscript as well.

> Once again, I repeat that "$\mathbf{R}$" should be replaced by "$\mathbb{R}$".

This has finally been addressed.

> Visibly my remark has been misunderstood. In my opinion, there are two possible ways of rigorously treating the additional information...
>
> ...in which case it should be clearly mentioned that P does not have the same meaning in Eqs. (1)-(6) and in Eq. (21);

It is both and neither of the points. The information contained in $P$ is always the same. It is everything that you (the agent performing inference) knows. The difference is what information you choose to ignore. We added a sentence to the description of the standard ETPF that almost everything in $P$ is typically ignored.

---

## Editor Decision (ED2)

Dear Colleagues,

Referee 1 (who is still Alban Farchi) has sent his evaluation of the latest version of your paper. He recommends acceptance subject to technical corrections, and makes one specific suggestion in that respect.

I myself as Editor have also a number of additional suggestions for corrections.

1. L. 51. It might be useful to mention at the start that the weights $w_j$ must be positive and must sum up to 1.

2. There seems to be an inconsistency of notation between the definition of $\mathbf{X}^f$ (l. 48) and later equations such as (2), (3), (5, first line) and others ($x^f_j$ or $\mathbf{X}^f_j$?). Please check.

3. L. 277, *All reported results are for differences that are two or more standard deviations apart*. The meaning is obscure.

4. L. 285 (and elsewhere) *trace-state normalized matrix*. Meaning ?

5. L. 158, do you mean … *with respect to* $\pi(X^f \,|\, P)$, or what ?

6. L. 148. Exponent of $\mathbf{R}$, $n \times N^f$

7. List of references. Reference to Popov *et al*. (2020) incomplete.

8. L. 15, *constraints* → *constrains*

9. L. 394, … *the authors' experience* … (plural, I presume)

10. L. 284, … *over*  *the whole manifold* …

Please make the necessary corrections (unless you disagree).

I thank you for having thought of *Nonlinear Processes in Geophysics* for publishing your paper.

---

## Author Response (AR3)

**Response to Referees**

Andrey A Popov et al.

May 26, 2022

**1  Response to Reviewer**

> As a final comment, I would suggest a small reformulation of the paragraph right before Section 5.6 : I am not sure to fully understand what is the difference between the "radius" and the "internal parameter".

We have added a citation clarifying the appearance of this internal parameter.

**2  Response to the Editor**

> L. 51. It might be useful to mention at the start that the weights wj must be positive and must sum up to 1.

We have added a clarification to this.

> There seems to be an inconsistency of notation between the definition of Xf (l. 48) and later equations such as (2), (3), (5, first line) and others (xf j or Xf j ?). Please check.

We have changed the definition to be better in line with the rest of the text.

> L. 277, All reported results are for differences that are two or more standard deviations apart. The meaning is obscure.

We have changed this to "All reported results are for statistically significant differences." We hope this aliviates any confusion.

> L. 285 (and elsewhere) trace-state normalized matrix. Meaning ?

We have added an equation to clarify this.

> . L. 158, do you mean ... with respect to $\pi(X^f|P)$, or what ?

Yes, this has been fixed.

> . L. 148. Exponent of R, $n \times N^f$

We have instead opted to say $N = N^f$ and $N^a = N + M$.

> List of references. Reference to Popov et al. (2020) incomplete.

We have re-exported the reference.

> L. 15, constraints $\rightarrow$ constrains

We have corrected this.

> L. 394, ... the authors' experience ... (plural, I presume)

This has been corrected.

> L. 284, ... over that of the whole manifold ...